# GC-MS- and NMR-Based Metabolomics and Molecular Docking Reveal the Potential Alpha-Glucosidase Inhibitors from *Psychotria malayana* Jack Leaves

**DOI:** 10.3390/ph14100978

**Published:** 2021-09-26

**Authors:** Tanzina Sharmin Nipun, Alfi Khatib, Zalikha Ibrahim, Qamar Uddin Ahmed, Irna Elina Redzwan, Riesta Primaharinastiti, Mohd Zuwairi Saiman, Raudah Fairuza, Tri Dewanti Widyaningsih, Mohamed F. AlAjmi, Shaden A. M. Khalifa, Hesham R. El-Seedi

**Affiliations:** 1Pharmacognosy Research Group, Department of Pharmaceutical Chemistry, Kulliyyah of Pharmacy, International Islamic University Malaysia, Kuantan 25200, Malaysia; tsn.np99@gmail.com (T.S.N.); zalikha@iium.edu.my (Z.I.); quahmed@iium.edu.my (Q.U.A.); elina@iium.edu.my (I.E.R.); 2Department of Pharmacy, Faculty of Biological Sciences, University of Chittagong, Chittagong 4331, Bangladesh; 3Faculty of Pharmacy, Airlangga University, Surabaya 60155, Indonesia; 4Institute of Biological Sciences, Faculty of Science, University of Malaya, Kuala Lumpur 50603, Malaysia; zuwairi@um.edu.my; 5Center for Research in Biotechnology for Agriculture (CEBAR), Faculty of Science, University of Malaya, Kuala Lumpur 50603, Malaysia; 6Faculty of Agricultural Technology, Brawijaya University, Malang 65145, Indonesia; dindafairuza@gmail.com (R.F.); tridewantiw@ub.ac.id (T.D.W.); 7Department of Pharmacognosy, College of Pharmacy, King Saud University, Riyadh 11451, Saudi Arabia; malajmii@ksu.edu.sa; 8Department of Molecular Biosciences, The Wenner-Gren Institute, Stockholm University, SE-106 91 Stockholm, Sweden; shaden.khalifa.2014@gmail.com; 9Division of Pharmacognosy, Department of Pharmaceutical Biosciences, Uppsala University, Biomedical Centre, P.O. Box 591, SE 751 24 Uppsala, Sweden; hesham.el-seedi@farmbio.uu.se; 10Department of Chemistry, Faculty of Science, Menoufia University, Shebin El-Kom 32512, Egypt; 11International Research Center for Food Nutrition and Safety, Jiangsu University, Zhenjiang 212013, China

**Keywords:** *P. malayana*, *α*-glucosidase, multivariate data analysis, GC-MS, NMR, molecular docking

## Abstract

*Psychotria malayana* Jack leaf, known in Indonesia as “daun salung”, is traditionally used for the treatment of diabetes and other diseases. Despite its potential, the phytochemical study related to its anti-diabetic activity is still lacking. Thus, this study aimed to identify putative inhibitors of α-glucosidase, a prominent enzyme contributing to diabetes type 2 in *P. malayana* leaf extract using gas chromatography-mass spectrometry (GC-MS)- and nuclear magnetic resonance (NMR)-based metabolomics, and to investigate the molecular interaction between those inhibitors and the enzyme through in silico approach. Twenty samples were extracted with different solvent ratios of methanol–water (0, 25, 50, 75, and 100% *v*/*v*). All extracts were tested on the alpha-glucosidase inhibition (AGI) assay and analyzed using GC-MS and NMR. Multivariate data analysis through a partial least square (PLS) and orthogonal partial square (OPLS) models were developed in order to correlate the metabolite profile and the bioactivity leading to the annotation of the putative bioactive compounds in the plant extracts. A total of ten putative bioactive compounds were identified and some of them reported in this plant for the first time, namely 1,3,5-benzenetriol (**1**); palmitic acid (**2**); cholesta-7,9(11)-diene-3-ol (**3**); 1-monopalmitin (**4**); *β*-tocopherol (**5**); *α*-tocopherol (**6**); 24-epicampesterol (**7**); stigmast-5-ene (**8**); 4-hydroxyphenylpyruvic acid (**10**); and glutamine (**11**). For the evaluation of the potential binding modes between the inhibitors and protein, the in silico study via molecular docking was performed where the crystal structure of *Saccharomyces cerevisiae* isomaltase (PDB code: 3A4A) was used. Ten amino acid residues, namely ASP352, HIE351, GLN182, ARG442, ASH215, SER311, ARG213, GLH277, GLN279, and PRO312 established hydrogen bond in the docked complex, as well as hydrophobic interaction of other amino acid residues with the putative compounds. The *α*-glucosidase inhibitors showed moderate to high binding affinities (−5.5 to −9.4 kcal/mol) towards the active site of the enzymatic protein, where compounds **3**, **5**, and **8** showed higher binding affinity compared to both quercetin and control ligand.

## 1. Introduction

Diabetes mellitus (DM) is one of the chronic diseases characterized by hyperglycemia in which blood sugar levels are dramatically elevated. It disrupts the body’s normal carbohydrate-, fat-, and protein-related metabolism and ultimately proves lethal if not properly treated or controlled [1]. It is a long-term disorder that has a massive effect on the lives and well-being of people, families, and communities around the world, and is also one of the leading causes of adult death. The prevalence of global diabetes in 2030 is expected to be 578 million (10.2%), rising to 700 million (10.9%) by 2045, which is very alarming [2]. Among three types of DM (type 1, type 2, and gestational DM), type 2 DM accounts for around 90% of all diabetes cases. Controlling plasma glucose level plays a significant role in inhibiting or avoiding type 2 DM. One treatment strategy to reduce postprandial hyperglycemia is the medication or diet to slow the synthesis or accumulation of glucose by preventing enzymes such as α-glucosidase that catalyze the hydrolysis of carbohydrates. Complex carbohydrates are hydrolyzed by one intestinal enzyme, namely *α*-glucosidases, into glucose and several other monosaccharides. The inhibitor of this enzyme can bind at the active or allosteric site to inhibit the enzymatic action, thus making the enzyme unable to catalyze the hydrolysis reaction of carbohydrates and reduce the blood glucose level [1,3].

The current treatment of diabetes relies on many synthetic medicines that are available in the market, but the long-term use of these drugs may implicate side effects [4]. Likewise, these drugs are costly for low-income people [5]. Considering the aforementioned hindrances, there is a growing concern in herbal-based remedies to combat DM [6]. One of the potential traditional herbs for the treatment of DM is *Psychotria malayana* Jack leaf. It is a medicinal plant which is known as “salung” or “loning” in Indonesia, and is found throughout southeast Asian countries. It has many applications in traditional medicine due to its potential pharmacological activities. The *Psychotria* species historically used in north Sumatra by Karo people to treat diabetes [7]. Despite its potential as an anti-diabetic agent, the phytochemical study to support this claim is still lacking. A major alkaloid, hodgkinsine and other minor compounds, namely calycanthine, (+/−)-chimonanthine, meso-chimonanthine, 2-ethyl-6-methylpyrazine, and 3-methyl-1,2,3,4-tetrahydro- γ -carboline, had been isolated from the leaves of this plant [8]. In spite of the evidence of analgesics, anti-bacterial, and convulsant activities [9,10,11], no report has been found on the anti-diabetic activity of the isolated compounds from this plant.

One of the effective approaches used to investigate the bioactive compounds possessing anti-diabetic activity in herbs is metabolomics. This holistic approach has recently become popular due to its ability to directly pinpoint the known bioactive compounds in the mixture of compounds, especially in the crude extract [12]. Furthermore, the bioactive compound’s bioactivity identified in this approach is well described and matched to that of the crude extract. Thus, it circumvents missing the bioactivity caused by sample fractionation, which is commonly experienced in bioassay-guided isolation. This approach employs a supervised multivariate data analysis (partial least square) correlating X-variables (instrumental signals from the metabolites) to Y-variables (bioactivity of the samples). The X-variables are situated close to the Y-variables in the loading plot, indicating the positive correlation to the bioactivity [13].

GC-MS is one of the analytical instruments commonly used in metabolomics. It has several advantages, such as high sensitivity, excellent resolution, and good reproducibility, which are necessary for the study of a complex biological mixture [1]. GC-MS-based metabolomics has been used for identifying the alpha-glucosidase inhibitors from various medicinal plants, including *Tetracera scandens* [12], *Clinacanthus nutans* [1], *Cosmos caudatus* [14], *Achras sapota* L. [15], and *Paederia foetida* L. [16]. ^1^H-NMR spectroscopy has been frequently used due to its ability to identify the metabolites over a broad range of dynamics with a single measurement, high accuracy, and easy sample preparations [17,18,19]. This approach has been applied to identify bioactive metabolites from *Phyllanthus niruri* extracts [20] and *Muntingia calabura* leaves ethanolic extract [18].

This study aims to identify *α*-glucosidase inhibitors in *P. malayana* leaf extract using GC-MS and NMR-based metabolomics, and investigate the molecular interaction between those inhibitors and the enzyme through in silico molecular docking. The purpose of docking is to accurately predict a ligand’s positioning within a protein binding pocket and estimate the binding strength via a docking score [21]. It is anticipated that the identified compounds from this study may serve as a basis in the development of *α*-glucosidase inhibitors.

## 2. Results

Several steps were applied in order to identify the compounds having AGI activity in *P. malayana* leaves extracts. The plant extracts were analyzed in parallel through GC-MS, NMR, and AGI activity test. Multivariate data analysis was used to pinpoint the GC-MS and NMR signals responsible for the AGI activity. The profiles of the selected signals were then compared to the available databases in order to identify the putative compounds in the plant extracts. The molecular docking study has been performed to support the findings from the GC-MS- and NMR- based metabolomics analysis.

### 2.1. α-Glucosidase Inhibition Assay

The percentage of *α*-glucosidase inhibition (AGI) of different extracts of the *P. malayana* leaves at the concentration of 5 µg/mL is demonstrated in Table 1. Pure methanol extracts showed the highest inhibition (71.7%) compared to other extracts, which was not significantly different (*p* > 0.05) from the positive control (quercetin). Conversely, 75% methanol extract exhibited the lowest (36.1%) inhibition against the enzyme, and no significant difference (*p* > 0.05) was noticed to that of 0% and 50% methanol extract. The bioactivity of the 25% methanol extracts (49.1%) exhibited a significantly different (*p* < 0.05) and higher compared to the 0% methanol extract. The bioactivity trend showed the following order: 100% > 25% > 50% ≈ 0% ≈ 75% methanol extract.

### 2.2. GC-MS-Based Metabolomics

#### 2.2.1. Multivariate Data Analysis

Partial Least Square (PLS) model was utilized to investigate the correlation between GC-MS data (X variables) and AGI activity (Y variable) of each plant extract. Pareto scaling was used to eliminate the GC-MS noise, thereby eliminating the bias results. The quality of fit is explained by the cumulative R^2^Y values, which indicate the percentage of variation explained by the model, and the cumulative Q^2^Y value, which is a variation that could be predicted by the model after the cross-validation. Figure 1A depicts the summary of fit based on the PLS model. Four principal components (PCs) were developed by the model fitting. The total variation explained was observed at 97.8%. The biggest variation in the samples was explained by PC 1 (54.7%). The PC 2, 3, and 4 explained 18.5, 19.1, and 5.5% of sample variations, respectively. The R^2^Y cumulative and Q^2^Y cumulative were found to be 0.98 and 0.89, respectively. It is confirmed to be credible based on the aforementioned standard. Model fitness and predictive capacity are considered credible if both the R^2^Y and Q^2^Y cumulative values are greater than 0.5, and the difference between both values is not bigger than 0.2 [13].

Another parameter for validation is the capability of the model to predict the y variable (bioactivity) based on the GC-MS data of the sample. The predicted value should be close to the actual bioactivity value expressed by the R^2^ value of the regression line between the observed versus predicted plot, as shown in Figure 1B. From the plot, it was found that all the points (samples) were located close to the regression line, which indicates the goodness of the model. The R^2^ value for this model was found to be 0.9776. The R^2^ value of more than 0.9 indicates the model is valid [13].

The Root Mean Square Error of Estimation (RMSEE) and Root Mean Square Error of Cross-Validation (RMSECV) values for this model were found at 0.0162011 and 0.0370577, respectively. RMSEE and RMSECV are also considerable parameters to measure the accuracy and performance of the model. RMSEE is an estimate of the model’s average deviation from the data. On the other hand, RMSECV is an indicator of the consistency of the model in predicting new samples. For new samples, the smaller the RMSECV value, the higher the predictive accuracy [1,13].

A score scatter plot based on the PLS model is shown in Figure 2A. The plot was aimed to observe separation among the samples. The most active extract (methanol extract) was located at the most positive side of the PLS component 1, whereas the least active (75% methanol–water) extracts were situated at the negative side. PLS component 2 could not differentiate the active extract from the non-active extract.

The loading scatter plot of the PLS model is shown in Figure 2B. This plot figures out the correlation between the AGI activity (Y variables) and the samples’ *m*/*z* value (X variables). The *m*/*z* value nearer to the AGI activity indicates a positive correlation to the bio-activity. The variable influence on projections (VIP) values of all above selected signals were more than 1 (Appendix A), and the jack-knifing error bars did not cross the zero (0) line of w*c(1) (Appendix A), indicating the significant correlation of these signals to the AGI activity. After comparing with the NIST 14 database based on their mass fragmentation (Appendix A) the signals were identified and shown in Figure 2B, while the chemical structures of the identified compounds are displayed in Figure 3.

#### 2.2.2. Bioactive Metabolites Profiling of *P. malayana* Extracts by GC-MS

The metabolites were identified based on the National Institute of Standards and Technology (NIST) 14 database. The fragment *m*/*z* spectra of each putative compound are shown in Appendix A. The metabolites with the similarity index (SI) of more than 90% can be acknowledged as the putative compounds [22,23]. Various groups of compounds possessing AGI activity, such as fatty acid (palmitic acid (**2**)), phenolics(1,3,5-benzenetriol (**1**), *β*-tocopherol (**5**), and *α*-tocopherol (**6**)), steroid (cholesta-7,9(11)-diene-3-ol (**3**), 24-epicampesterol (**7**), and stigmast-5-ene (**8**)), and glyceride (1-monopalmitin (**4**)) were identified. Apart from these bioactive metabolites, one non-active metabolite, namely, myo-inositol (**9**), was also identified. Figure 4 depicts all the identified bioactive compounds, which are labeled at the respective peaks in the representative chromatogram of the most active 100% methanol extract. The details of the identified compounds are mentioned in Table 2. Among them, cholesta-7,9(11)-diene-3-ol (**3**) exhibited highest % peak area (3.42%), followed by 1,3,5-benzenetriol (**1**) (0.61%), while the % peak area of the remaining compounds was less than 1%.

#### 2.2.3. Bioactive Confirmation of Three Pure Bioactive Compounds

Table 3 depicts the IC_50_ value of palmitic acid, methanol extract (most active extract against AG), and quercetin (positive control). The IC_50_ value of palmitic acid was found significantly different (*p* < 0.05) and higher compared to methanol extracts and quercetin, but showed potential inhibitory activity against AG enzyme. This finding was found to be in line3 with the previous research study where palmitic acid exhibited potential AGI activity as a pure compound [1].

On the other hand, 1-monopalmitin and *α*-tocopherol failed to show their inhibitory activity against AG enzyme as pure compounds. However, along with the methanol extracts of *P. malayana* leaves, both of these compounds showed synergistic activity. Table 4 and Table 5 present the synergistic activity of 1-monopalmitin and *α*-tocopherol, respectively. From the result, it was observed that 1-monopalmitin as a pure compound showed no activity (0%) at all concentrations (4, 2, 1, 0.5, and 0.25 μg/mL) against AG enzyme. However, the four mixtures (4 μg/mL 1-monopalmitin + 2 μg/mL methanol extract, 2 μg/mL 1-monopalmitin + 2 μg/mL methanol extract, 1 μg/mL 1-monopalmitin + 2 μg/mL methanol extract, and 0.5 μg/mL 1-monopalmitin + 2 μg/mL methanol extract) exhibited significantly (*p* < 0.05) higher percentage of AGI (75.59, 61.89, 59.03, and 41.14%, respectively) compared to the pure compound (0% at 4, 2, 1, 0.5, and 0.25 μg/mL of 1-monopalmitin) and methanol extract (31.84% at 2 μg/mL).

Furthermore, as a pure compound, *α*-tocopherol exhibited significantly (*p* < 0.05) lower percentage of inhibition (10.93, 7.84, 6.07, 3.61, and 0%) against AG enzyme at all concentrations (4, 2, 1, 0.5, and 0.25 μg/mL, respectively) compared to both mixture and extract. In contrast, all five of the mixtures (4 μg/mL *α*-tocopherol + 2 μg/mL methanol extract, 2 μg/mL *α*-tocopherol + 2 μg/mL methanol extract, 1 μg/mL *α*-tocopherol + 2 μg/mL methanol extract, 0.5 μg/mL *α*-tocopherol + 2 μg/mL methanol extract, and 0.25 μg/mL *α*-tocopherol + 2 μg/mL methanol extract) showed significantly higher % of AGI with the value of 97.08, 90.94, 85.22, 74.55, and 56.99%, respectively, compared to the pure *α*-tocopherol and methanol extracts (31.18% at 2 μg/mL).

### 2.3. NMR-Based Metabolomics

#### 2.3.1. Multivariate Data Analysis

Orthogonal Partial Least Square (OPLS) model was utilized to investigate the correlation between NMR data (x variables) and AGI activity (y variable) of each plant extract. Pareto scaling was used to eliminate the NMR noise, thereby eliminating the bias results. The quality of fit is explained by the cumulative R^2^Y values, which indicate the percentage of variation explained by the model, and the cumulative Q^2^Y value, which is a variation that could be predicted by the model after the cross-validation. Figure 5A depicts the summary of fit based on the OPLS model. Six principal components (PCs) were developed by the model fitting. The total variation explained was observed at 95.6%. The biggest variation in the samples was explained by PC1 (36.7%). Apart from this, the PC2, PC3, PC4, PC5, and PC6 explained 19.5, 23.9, 7.6, 4.6, and 3.3% of sample variations, respectively. The R^2^Y cumulative and Q^2^Y cumulative were found to be 0.96 and 0.77, respectively. It is confirmed to be credible based on the aforementioned standard. Model fitness and predictive capacity are considered credible if both the R^2^Y and Q^2^Y cumulative values are greater than 0.5, and the difference between both values is not bigger than 0.2 [13].

Another parameter for validation is the capability of the model to predict the y variable (bioactivity) based on the NMR data of the sample. The predicted value should be close to the actual bioactivity value expressed by the R^2^ value of the regression line between the observed versus predicted plot, as shown in Figure 5B. From the plot, it was found that all the points (samples) were located close to the regression line, which indicates the goodness of the model. The R^2^ value for this model was found to be 0.9562. The R^2^ value of more than 0.9 indicates the model is valid [13].

The Root Mean Square Error of Estimation (RMSEE) and Root Mean Square Error of Cross-Validation (RMSECV) values for this model were found at 0.0243207 and 0.0532356, respectively. RMSEE and RMSECV are also considerable parameters to measure the accuracy and performance of the model. RMSEE is an estimate of the model’s average deviation from the data. On the other hand, RMSECV is an indicator of the consistency of the model in predicting new samples. For new samples, the smaller the RMSECV value, the higher the predictive accuracy [1,13].

A score scatter plot based on the OPLS model is shown in Figure 6A. The plot was aimed to observe separation among the samples. The most active extracts (100% methanol–water) of *P. malayana* leaves were located at the most positive side of the OPLS component 1, whereas the least active (75% methanol–water) extracts were situated at the negative side.

The loading column plot of the OPLS model is shown in Figure 6B. This plot figures out the correlation between AGI activity (Y variables) and the samples’ *δ* value (X variables). Jack-knifing error bars were calculated to check the significant correlation between AGI activity and chemical shift. The error bars that did not cross the 0 line of the *y*-axis indicate the respective chemical shifts were significantly correlated to AGI activity (per IC_50_) and vice versa [13,24]. It was further confirmed that all of the selected signals showed the variable influence on projections (VIP) value more than 1, indicating the significant correlation (Appendix A). Although many *δ* values showed the positive correlation to AGI activity (per IC_50_), after comparing with the available databases and published reports, only two (compounds **10** and **11**) could be identified and labeled in this plot. Besides this, another four non-active compounds (compounds **9**, **12**, **13**, and **14**) were also labeled where the error bars crossed the 0 line of the *y*-axis. Among these four, compound **9** was also detected in GC-MS-based metabolomics as a non-active compound. The chemical structures of the metabolites are displayed in Figure 7.

#### 2.3.2. Identification of Putative Bioactive and Other Metabolites

Identification of two putative bioactive compounds, namely 4-hydroxyphenylpyruvic acid (**10**) and glutamine (**11**), was performed by analyzing ^1^H-NMR and *J*-resolved. Figure 8 shows the 2D *J*-resolved and Table 6 depicts the ^1^H-NMR chemical shifts, proton number, type of signals, and coupling constants of the identified bioactive (**10** and **11**) and four non-active putative metabolites, namely myo-inositol (**9**), sucrose (**12**), *β*-glucose (**13**), and *α*-glucose (**14**) in *P. malayana* leaves extract. Compound **9** (myo-inositol) had also been detected in GC-MS-based metabolomics analysis as a non-active compound. Compound **10** showed two doublets (H-3′; H-5′, and H-2′; H-6′) and one singlet (H-3) at δ 6.78, δ 7.11, and δ 4.03, respectively, which were confirmed by the *J*-resolved spectrum (Figure 8A,B). Compound **11** exhibited two multiplets at δ 2.44 and δ 2.12 that were assigned to H-4 and H-3, respectively, and were determined by the *J*-resolved spectrum (Figure 8C). For compound **9**, one doublet of doublets (10.0 Hz; 4.0 Hz) and one triplet (9.0 Hz) were found at δ 3.51 (H-1) and δ 3.20 (H-5), respectively and were confirmed by *J*-resolved spectrum (Figure 8B). One doublet (4.0 Hz) was assigned to H-2 (glucose moiety) of compound **12** at δ 5.40 was confirmed by *J*-resolved spectrum (Figure 8B). Compound **13** exhibited one doublet (8.0 Hz) at δ 4.58 that was assigned to H-2, and compound **14** also displayed one doublet (4.0 Hz) at δ 5.18 that was assigned to H-2. Identification of both of glucose were verified by *J*-resolved spectrum (Figure 8B).

### 2.4. Molecular Docking Study of Putative Compounds Identified by GC-MS and NMR Analysis

The ten compounds identified using GC-MS- and NMR-based metabolomics were investigated using molecular docking to determine the possible binding mode that can explain their inhibition activities. Table 7 depicts the values of binding affinity of the ten tentative compounds along with the positive control (quercetin) and the control ligand (ADG) towards AG enzymatic protein (PDB ID: 3A4A). In order to validate the docking parameters, a control docking procedure was conducted utilizing ADG. On the other hand, a comparison with the identified metabolites was carried out with quercetin. The compound–enzyme docked complex is considered the best-docked if it exerts the most negative value, reflecting the strong binding affinity of the docked complex. It was observed that the re-docked ADG binds with the enzyme (PDB ID: 3A4A) in a way similar to its crystallographic configuration. The root mean square deviation (RMSD) value of the re-docked ADG was noted at 0.633 Å, implying that the selected docking parameters are capable of reproducing the crystallized conformation and are also considered to be acceptable as the value is less than 1.5 Å [37]. In total, seven amino acid residues of the enzyme, including ASP352, GLH277, ASH215, HIE112, ASH69, ARG442, and HIE351 were involved in hydrogen bond interactions with the control ligand. Meanwhile, ASH215 and GLH277 were interacted via hydrogen bonds, and PHE303, ASP352, and ARG442 were interacted via other interactions in the docked complex involving quercetin.

From the docking result, it was found that among the identified compounds, compound **1** exhibited the lowest, and compound **8** showed the highest binding affinity towards the enzyme. The seven compounds (**2, 3, 4, 5, 6, 7, 8,** and **10**), except compounds **1** and **11**, showed higher binding affinities towards the enzyme compared to ADG (−6.0 kcal/mol). In addition, compounds **3, 5,** and **8** exhibited the greater binding affinity of −9.1, −8.6, and −9.4 kcal/mol, respectively, when compared to that of quercetin (−8.4 kcal/mol). On the other hand, compounds **1**, **4**, **6**, **7,** and **11** (−5.5, −6.1, −7.9, −7.7, and −5.8 kcal/mol, respectively) showed lower binding affinities than that of quercetin.

Figure 9 depicts the 3D superimposed diagram rendered by Pymol that explains the simulated binding site of all the ten bioactive compounds along with ADG and quercetin on the enzyme (3A4A). The figure indicates that all compounds bind to domain A of AG enzyme, where all the catalytic residues are present. ADG and quercetin were also found to bind at the same site, implying that the detected compounds might follow a similar inhibition mechanism.

Appendix A depicts the type of bond, bond distance (Å), and amino acid residues involved in the binding interactions of ten identified compounds, while Figure 10 shows the two-dimensional (2D) binding interactions between the identified compounds and the enzyme. Several interactions were observed in the docked complexes, including hydrogen bond, pi-sigma, pi-alkyl, and alkyl interactions. From the docking results, it was found that in the docked complex containing compound **1**, ASP352 (2.35 Å) and HIE351 (2.40 Å) interacted with the ortho positioned hydroxyl moiety, and GLN182 (2.70 Å) interacted with para positioned hydroxyl moiety via hydrogen bonds. The hydrogen bond formed between ASP352 and compound **1** is the strongest one. On the other hand, ARG442 and VAL216 exhibited pi-cation (4.18 Å) and pi-alkyl (5.47 Å) interactions, respectively, with the aromatic moiety of compound **1**.

In the compound **2**–AG docked complex, two hydrogen bonds were established by the protonated ASP215 (2.30 Å) and HIE351 (2.20 Å) with the hydroxyl moiety, while the third hydrogen bond formed by ARG442 (2.89 Å) with the carbonyl moiety of compound **2**. Among these three hydrogen bonds, the one involving HIE351 was predicted to be the strongest. Apart from this, TYR158 (4.91 Å) and PHE303 (4.66 and 5.21 Å) of the enzyme interacted with the aliphatic moiety of compound **2** via three pi-alkyl interactions.

From the docked complex containing compound **3**, only one hydrogen was observed involving SER311 and the hydroxyl moiety, with a bond distance of 2.20 Å. Three amino acid residues, namely TYR72 (4.61 Å), PHE178 (4.68 Å), and PHE303 (4.66 Å) interacted with the aliphatic moiety of compound **3** through pi-alkyl interactions, whereas TYR158 (5.26 Å) and HIE280 (5.25 Å) formed pi-alkyl and ARG215 formed two alkyl interactions (4.12 Å and 4.69 Å) with the alicyclic moiety of compound **3**. In addition, TYR72 and PHE178 interacted with the aliphatic moiety via pi-sigma interactions at the bond distances of 3.88 Å and 3.67 Å, respectively.

In the case of the compound **4**-3A4A docked complex, three hydrogen bonds were formed by ARG213 (2.89 Å), protonated GLH277 (2.90 Å), and ASP352 (2.87 Å) with the hydroxyl moiety, and the fourth hydrogen bond was established by GLN279 (2.92 Å) with the carbonyl moiety of compound **4**. ASP352 produced the strongest hydrogen bond with the smallest distance compared to others. Meanwhile, PHE303 (5.18 Å), HIE280 (5.17 Å), and TYR158 (4.79 and 5.21 Å) showed four pi-alkyl interactions with the aliphatic moiety of compound **4**, whereas ARG315 exhibited alkyl interaction at a bond distance of 4.54 Å with the aliphatic moiety of compound **4**.

The compound **5**–AG enzyme docked complex showed no hydrogen bond interaction. One pi-sigma (3.98 Å) and three pi-alkyl interactions (5.36, 5.00, and 4.77 Å) were formed by PHE303 with the aliphatic moiety of compound **5**. Moreover, PHE159 (4.75 Å), TYR158 (5.01 Å), PHE178 (4.23 and 5.35 Å), and HIE280 (4.72 Å) were also involved interactions via pi-alkyl bond with the aliphatic moiety of compound **5**. Additionally, two pi-sigma interactions with the bond distances of 3.97 and 3.86 Å were established by TYR72 and PHE178, respectively, with the aliphatic moiety. On the other hand, the aromatic moiety interacted with ARG315 via pi-alkyl (4.29 Å) and alkyl (4.16 Å) interactions. From the findings, it was observed that the long-chain aliphatic moiety of compound **5** established eleven hydrophobic interactions, whereas aromatic moiety involved only two hydrophobic interactions with the enzymatic protein.

The docked complex containing compound **6** exhibited only one hydrogen bond interaction with PRO312 at a distance of 2.92 Å with the hydroxyl moiety of compound **6**. The aromatic moiety of compound **6** interacted with HIE280 (4.38 and 4.79 Å) and PHE303 (5.07 Å) via pi-alkyl interactions and with ARG315 (4.76 Å) via alkyl interaction. Furthermore, ARG315 also established alkyl and pi-sigma interactions with the tetrahydropyran and aromatic moiety of compound **6** at distances of 4.15 Å and 2.97 Å, respectively. Furthermore, pi-sigma bonds were formed by TYR72 (3.38 and 3.78 Å) and PHE303 (3.87 Å) with the aliphatic moiety of compound **6**. Again, TYR72 and PHE303 were interacted with the aliphatic moiety via pi-alkyl interactions at the bond distances of 4.01 and 4.98 Å, respectively. Additionally, HIE351 (5.17 Å) and PHE178 (4.56 Å) were also exhibited pi-alkyl, whereas alkyl interaction was formed by valine 216 (4.77 Å) with the aliphatic moiety of compound **6**. From the results, it was found that though compound **6**-enzyme docked complex produced 14 hydrophobic and one hydrogen bond interaction, no catalytic residues of the enzyme took part in the interaction with this compound.

In the compound **7**-3A4A docked complex, the aliphatic moiety of compound **7** interacted with three amino acid residues, including PHE178 (4.65 Å), PHE303 (4.13 Å), and TYR72 (4.85 Å), via three pi-alkyl interactions. Besides pi-alkyl interaction, PHE178 also interacted with aliphatic moiety via pi-sigma interaction at a bond distance of 3.76 Å. The alicyclic moiety of compound **7** interacted with ARG315 and TYR158 through two alkyl (4.68 and 4.46 Å) and two pi-alkyl interactions (4.05 and 5.08 Å), respectively.

There was no hydrogen bond observed in the compound **8**-enzyme docked complex. Eight amino acid residues interacted with compound **8** via three types of interactions (pi-sigma, alkyl, and pi-alkyl interactions). Pi-sigma interactions were produced by TYR72 (3.82 and 3.80 Å) and PHE178 (3.71 Å) with the aliphatic moiety of compound **8**, whereas pi-alkyl interactions were established by TYR158 (4.35 and 5.33 Å), HIE280 (4.93 Å), and PHE303 (5.46 Å) with the alicyclic moiety of compound **8**. Besides pi-sigma, PHE178 also interacted with the aliphatic moiety via pi-alkyl interaction at a bond distance of 4.05 Å. The alicyclic moiety also interacted via three alkyl interactions with ARG315 at distances of 4.19, 4.46, and 5.03 Å. Furthermore, PHE303, PHE159, HIE351, and PHE178 established pi-alkyl interactions with the aliphatic moiety of compound **8** at the distances of 4.62, 5.43, 5.08, and 4.05 Å, respectively.

From the docking results, it was found that in the compound **10**–AG docked complex, ASH215 interacted with the aliphatic hydroxyl moiety of compound **10** via hydrogen bond with a distance of 2.51 Å. On the other hand, ARG442 exhibited hydrogen bond (2.19 Å) and pi-cation (4.69 Å) interactions with the carbonyl (positioned at carbon number 2) and aromatic moieties, respectively. The aromatic moiety also interacted with ASP352 via pi-anion interaction at a bond distance of 4.95 Å. The other carbonyl moiety (positioned at carbon number 1) formed a hydrogen bond with HIE351 residue at a distance of 2.14 Å.

In the compound **11**–AG docked complex, all residues were observed to interact via hydrogen bond. The first two hydrogen bonds were formed between HIE351 and the carbonyl moiety (positioned at carbon number 1, with distance 1.96 Å), also between ARG442 residue and another carbonyl moiety (positioned at carbon number 5, with distance 2.18 Å). The next hydrogen bond was formed between amino moiety (positioned at carbon number 2) and ASH215 (2.22 Å). The next two hydrogen bonds involved another amino moiety (positioned at carbon number 5) with ASP352 and GLH277 at the bond distance of 2.73 Å and 2.12 Å, respectively. The last two hydrogen bonds also involved ASP352 and GLH277 residues, with the hydroxyl moiety of compound **11** at the distance of 2.67 Å and 2.46 Å, respectively.

## 3. Discussion

### 3.1. AGI Activity

The finding of this study was in line with previous research in which AGI activity was found in the plant extract obtained using aqueous methanolic solvent during the extraction process. This solvent has been used previously to obtain *Tetracera scandens*, *Psiadia punctulata*, and *Cornus capitata* Wall leaves extracts with AGI activity [12,38,39]. The difference in the AGI activity among the extracts was explained by the difference in the metabolite profiles present in the extracts that were influenced by the solvent polarity during the extraction [40,41]. Quercetin was chosen as a positive control in this study as it is a potent AGI comparable to widely prescribed α-glucosidase inhibitors, such as acarbose, miglitol, and voglibose. In addition, unlike the others, quercetin is found widely in plants [42,43,44,45,46].

### 3.2. Putative Compounds Identified by GC-MS

Eight bioactive metabolites associated with *α*-glucosidase inhibitory activities of the *P. malayana* leaves were identified by GC-MS analysis, as shown in Figure 3. The occurrence of these compounds in this plant was reported for the first time in this study. The presence of these compounds was reported in other plants. Compounds **2**, **4**, and **6** have been identified as putative compounds and reported as *α*-glucosidase inhibitors which were identified from the methanolic leaves extracts of *Tetracera scandens* [12], while Murugesu et al. [1] and Alam et al. [47] had identified compounds **2, 4, 5,** and **8** as the putative *α*-glucosidase inhibitors from *Clinacanthus nutans* leaves. Compound **6** was detected as a putative compound in *Cosmos caudatus* ethanolic leaves extracts and proved effective as an *α*-glucosidase inhibitory agent [14,48]. In addition, Yumna et al. [49] had reported compound **7** as a potential anti-diabetic compound, detected from 70% ethanolic extract of *Sansevieria trifasciata* leaves. Compound **1** is a phenolic compound and also known as phloroglucinol. It was identified as a major compound in *Mukia maderaspatana* and *Ecklonia cava* and reported to have potential anti-diabetic activity by inhibiting hepatic gluconeogenesis in rat and mouse liver, respectively [50,51]. On the other hand, Akhtar et al. [52] had a novel olefinic rearrangement where compound **3** was synthesized and converted into cholesterol. The AGI activity of this compound is reported for the first time in this study.

Palmitic acid (compound **2**), as a pure compound, showed potential inhibitory activity against AG enzyme, which was in line with the previous research study in which palmitic acid had also been reported to exhibit potential AGI activity as a pure compound [1]. In contrast, both 1-monopalmitin (compound **4**) and *α*-tocopherol (compound **6**) were found inactive against AG enzyme as pure compounds. No report has been found that compounds **4** and **6** have AGI activity in the form of pure compounds. The present study revealed that both compounds **4** and **6** exhibited synergistic activity; alone as pure compounds, they were unable to exert the inhibitory activity against AG enzyme, but within the extracts, they showed significant inhibition due to the synergistic effect with other compounds present in the plant extracts. This finding was found to be in line with the previous research study wherein the isolated pure compounds (*β*-sitosterol and stigmasterol) from the hypoglycaemic fraction of *Parkia speciosa* seed showed no activity. Interestingly, the fraction contained only those two compounds. It indicated that the significant hypoglycaemic activity of *P. speciosa* was exhibited due to the synergistic action of *β*-sitosterol and stigmasterol [53]. Purification or isolation of compounds from the biologically active plant extracts may reduce the bioactivity of the pure compounds in most of cases [54]. Plant extracts generally consist of hundreds of thousands of metabolites, and synergism between a number of metabolites is possible as a consequence of the bioactivity of plant extracts [55]. To our knowledge, the synergetic activity of these two compounds with other phytoconstituents present in *P. malayana* leaves extracts has not been documented to date.

### 3.3. Putative Compounds Identified by NMR

From NMR analysis, the loading column plot obtained using OPLS (Figure 6B) pinpointed two putative compounds (**10** and **11**) correlating to AGI activity of *P. malayana* leaves extracts. The presence of these compounds in this plant had not been recorded elsewhere; thus, this study reports it for the first time. Compound **10** is a phenolic acid; Jindra et al. [28] had reported the presence of compound **10** in *Papavar somniferum* plants. Furthermore, Hou et al. [29] had identified this compound as a metabolite in the metabolic pathways of *Salvia miltiorrhiza* Bunge (Chinese medicinal plant) and *Salvia castanea* f. *tomentosa* Stib, where they used methyl jasmoate (phytohormone) as an inducer to synthesize phenolic acids in the hairy roots of *S. miltiorrhiza* Bange and *S. castanea* f. *tomentosa* Stib and quantified compound **10** using ^1^H-NMR. They found the chemical shifts of compound **10** around δ 6.00, δ 7.00, and δ 4.00, which are in line with the present study. Similar chemical shifts had also been reported by the Humane Metabolome Database (HMDB ID: HMDB0000707). *S. miltiorrhiza* Bange is a popular Chinese medicinal herb that is extensively used for the management of cardiovascular and cerebrovascular diseases. The major constituents of this herb are phenolic acids. *S. castanea* f. *tomentosa* Stib possess similar pharmacological activities to *S. miltiorrhiza* Bange. Though no anti-diabetic activity has been reported for the compound **10**, a significant quantities of this compound was reported in methyl jasmoate treated hairy roots of *S. miltiorrhiza* Bange and *S. castanea* f. *tomentosa* Stib [29,56]. On the other hand, phenolic acids are highly potential for various biological activities, including antioxidant and antimicrobial activities [57].

Compound **11** is a non-essential amino acid, which can be synthesized by the human body and also widely available in nature. This amino acid had been identified in numerous medicinal plants, including *Panax ginseng* C.A. Meyer (seed, root, stem, leaves, and whole plant) [30], *Solanum tuberosum* L. (six potatoes cultivars) [31], *Clinacanthus nutans* (leaf and stem) [32], *Nicotiana tabacum* (leaves) [26], and *Hilliardiella elaeagnoides* (leaves and stems) [33]. Furthermore, this amino acid had also been identified in *Coprinus comatus* (medicinal mushrooms) [34]. Wang et al. [58] had reported the hypoglycemic effect of this amino acid, where skeletal muscle L6-cells were used. From that study, it was revealed that compound **11** increased the hypoglycemic effect of insulin through insulin signaling and glycogen synthesis pathways. Besides this, to control the blood glucose level of T2DM patients, glutamine supplements are used [59,60]. The ^1^H-NMR chemical shifts displayed by this amino acid are in line with the several reported research studies, where two multiplets were found around 2.10 and 2.40 ppm [25,26,27,32,61,62,63].

Besides these two tentative bioactive compounds (**10** and **11**), another four metabolites (**9** and **12**–**14**) were also identified in the *P. malayana* leaves extracts through ^1^H-NMR. All four of these compounds are widely available in various plant species. Choi et al. [26] applied an NMR-based metabolomics approach to identify the metabolites present in *Nicotiana tabacum* leaves and detected compound **9** as one of the metabolites. The chemical shifts shown by compound **9** are in line with the previous studies, where NMR-based metabolomics analysis was used to identify this compound [25,26,27]. Compounds **12, 13,** and **14** are three types of sugars, which are very common and abundant in plant leaves. Plenty of research studies had performed NMR-based metabolomics analysis and reported the presence of these sugars in various plant species. The present study detected these sugars in ^1^H-NMR around 4.00 and 5.00 ppm, which are in line with the previously reported studies [26,32,35,36].

### 3.4. In-Silico Study

The best way to validate the AGI activity of these compounds is through testing each pure compound by in vitro analysis. However, the limited concentration of these compounds in this plant makes it hard to isolate. In addition, some compounds are not available commercially. For this reason, the binding characteristics between the identified compounds and the enzyme’s active site have been evaluated using in silico molecular docking.

There are 589 amino acids in the 3A4A protein. Three domains are involved in this protein. Domain A consists of 1-113 and 190-512 amino acids, whereas domain B and C are made up of 114-189 and 513-589 amino acids, respectively [64]. The edge of the C-terminal of domain A contains three catalytic residues (ASP215, GLU277, and ASP352) [64,65]. The enzyme’s active site is represented by the catalytic residues. As ADG is a co-crystallized ligand/substrate, so the residues involved in the ADG–AG docked complex, were indicating the actual catalytic residues as well [65].

The findings from the in silico analysis of the compounds identified by GC-MS analysis showed that all putative metabolites, except compound **1,** have moderate to good affinities towards the enzyme’s active site, indicating the ability to bind, slow down the catalytic reaction, and eventually inhibit the enzyme. The binding affinity of compound **1** is lower than the ADG, which may suggest the possibility of synergism effect of this compound with other compounds in the plant in order to exhibit the AGI activity. Unfortunately, the synergism effect could not be examined through the in silico molecular docking in this study due to a technical limitation. Further investigation in this regard may be needed to confirm this hypothesis in the future. Among the five amino acid residues involved in compound **1**–AG docked complex, three residues (ASP352, HIE351, and ARG442) also showed interaction in the ADG–AG complex. Moreover, one catalytic residue (ASP352) was also involved in the compound **1**–AG docked complex via hydrogen bond. Though the binding affinity of compound **1** towards AG enzyme was lower than both ADG and quercetin (known inhibitor), the number of enzymatic amino acid residues involved in hydrogen bond interactions with this compound was high, which may contribute to the AGI activity [65]. In addition, there are three hydroxyl groups present in the compound **1**–AG docked complex, which also play a vital role in interacting with the hydrophilic amino acid residues by forming hydrogen bonds that contribute to the inhibitory activity [66,67].

Among three pure compounds (compound **2**, **4**, and **6**), compound **2** showed promising in vitro AGI activity with the IC_50_ value of 8.04 μg/mL along with good biding affinity values towards the enzyme in the docking study. In both in vitro and in silico studies, compound **2**, palmitic acid, exhibited lower AGI activity (IC_50_ = 8.05 μg/mL) and binding affinity-value (−6.1 kcal/mol) compared to quercetin (IC_50_ = 1.86 μg/mL; −8.4 kcal/mol, respectively), which is in line with the previous study [1]. Furthermore, Nokhala et al. [12] also reported a similar binding affinity value of palmitic acid towards the enzyme. Compound **2** showed more hydrogen bond interactions and less hydrophobic interactions towards the enzyme compared to that of the quercetin–AG docked complex. The free hydroxyl groups of compound **2** provided the additional sites to produce hydrogen bond with the amino acid residues of the enzyme [12]. In addition, three (HIE351, ASH215, and ARG442) amino acid residues that are present in the ADG–AG docked complex, were also involved in the compound **2**–AG docked complex via hydrogen bond interaction, indicating good binding affinity towards the active site of the enzyme [1,68]. Furthermore, one (ASH215) catalytic residue took part in the interaction of compound **2**–AG docked complex, indicating the good inhibitory activity against the enzyme. On the other hand, though compounds **2** and **4** exhibited the same binding affinity values (−6.1 kcal/mol), the result from the in vitro experiment showed that compound **4** did not exhibit the bioactivity as an individual compound. However, the finding is described in Section 2.2.3, showed that this compound has synergistic activity to produce AGI activity. In addition, compound **6** also exhibited synergistic activity in the bioassay, whereas in docking study, it showed higher binding affinity (−7.9 kcal/mol) towards the enzyme compared to ADG (−6.0 kcal/mol) and lower binding affinity compared to quercetin. The synergistic effect of the ligand cannot be investigated in docking as the ligands were docked to enzymatic protein individually. Chen [69] highlighted that the docking study alone would not be able to provide the most accurate findings. As there is a possibility of false-negative and false-positive results, verification via in vitro experiment is recommended to support the docking findings [69,70,71].

On the other hand, the binding affinity showed by compound **7** (−7.7 kcal/mol) towards the enzyme was found higher than ADG but lower than that of quercetin. The docked complex containing compound **7** exhibited more hydrophobic interactions compared to that of both ADG–AG and quercetin–AG docked complexes. A higher number of hydrophobic interactions was observed due to the presence of extra methyl group of this compound [12]. However, the amino acid residues that took part in ADG–AG docked complex, were absent and did not participate in the hydrogen bond interaction of the compound **7**–AG. As there were no catalytic residues involved in the docked complex containing compound **7,** this might be indicating the possibility of indirect inhibition of this compound towards the enzyme [1,68].

Compounds **3**, **5**, and **8** exhibited higher binding affinity compared to both quercetin and ADG, indicating the greater inhibition activity against the enzyme. There were very few hydrogen bonds involved in the docked complexes of these compounds to the enzyme active site. The interaction of the compounds to the enzyme was dominantly linked through hydrophobic contact in order to produce an interactive inhibition. This is in line with a finding reported by Nokhala et al. [12]. Our result showed that no catalytic residues were involved in these docked complexes, suggesting these compounds might follow an indirect binding mechanism to inhibit the catalytic reaction of the enzyme, as mentioned in the previous research [1].

From the docking findings of the compounds identified by NMR analysis, it was observed that both putative compounds (compound **10** and **11**) showed lower value of binding affinities towards the active site of the enzyme compared to that of known competitive inhibitor (quercetin), indicating the compounds’ moderate (−5.8 and −6.5 kcal/mol) inhibition activity against the enzyme. Among these two compounds, compound **10** showed higher binding affinities compared to ADG (−6.0 kcal/mol). Four (ASP352, protonated ASH215, ARG442, and HIE351) catalytic residues were involved in compound **10**–AG docked complex. Though the binding affinity of this compound was lower than quercetin (−8.4kcal/mol), the presence of catalytic amino acid residues in the molecular binding interactions may contribute to the inhibition activities of this compound [65]. Moreover, the carbonyl and hydroxyl moieties of this compound involved in the strong hydrogen bond interactions with the catalytic residues, indicating the key role of these functional groups to stabilize the docked complex and contributing to the AGI activity [67]. The binding affinity value of this compound (−6.5 kcal/mol), indicates the moderate affinity towards the enzyme [12].

On the other hand, compound **11** showed lower binding affinities (−5.8 kcal/mol) towards the enzyme compared to that of both quercetin and ADG. A total of seven hydrogen bonds were established in the docked complex of this compound. Five catalytic residues (protonated ASH215, ARG442, ASP352, GLH277, and HIE351) were involved in the compound **11**–AG docked complex. The functional groups, namely hydroxyl, carbonyl, and amino moieties, of this compound actively participated in the interactions with the active site residues of the enzyme, and may have contributed to the inhibition activities. The findings are in line with the previous study, where it had been reported that the oxygen and nitrogen atoms played an important role in exerting the inhibition activities by forming hydrogen bonds with the residues of the enzyme [72,73]. On the other hand, the lower binding affinity exhibited by the compound **11**–AG docked complex may be due to the absence of hydrophobic interactions that may also play a crucial role in ligand binding towards the AG enzyme. Along with the hydrogen bond, hydrophobic interaction also plays a vital role in stabilizing the docked complex and enhancing the binding affinity of the ligand at the binding interface [74].

## 4. Materials and Methods

### 4.1. Materials

Pyridine anhydrous, methanol, and alpha-glucosidase (from Saccharomyces cerevisiae) were procured from Sigma-Aldrich (St. Luis, MO, USA), Merck (Darmstadt Germany), and Megazyme (Wicklow, Ireland), respectively. Potassium dihydrogen phosphate, glycine, dimethyl sulphoxide, methanol-*d*4 (CH_3_OH-*d*_4_), and deuterium oxide (D_2_O) were purchased from Merck (Darmstadt Germany), while *N*-methyl-*N*-(trimethylsilyl) trifluoroacetamide (MSTFA), methoxamine hydrochloride, quercetin (standard), p-nitrophenyl-α-D-glucopyranoside (PNPG), and trimethylsilylpropanoic acid were obtained from Sigma-Aldrich (St. Louis, MO, USA).

### 4.2. Plant Material

The *P. malayana* Jack leaves were obtained from Cermin at Sarolangun, Jambi, Indonesia, and identified by Shamsul Khamis (botanist at Universiti Putra Malaysia). The leaves were washed with water to remove debris and dried at room temperature (27 ± 1 °C) for 7 days. The plant sample was deposited to Herbarium at Kulliayyah of Pharmacy, International Islamic University Malaysia, Kuantan Campus, with the voucher specimen number PIIUM 008-2.

### 4.3. Plant Extract Preparation

The washed leaves were dried at ambient temperature and ground to a coarse powder using a grinder. The powdered leaves were stored at −80 °C [1]. Various methanol–water ratios (0%, 25%, 50%, 75%, and 100%) were used as an extraction solvent to prepare twenty *P. malayana* leaves extracts (four technical replicates, *n* = 4, in each group). Extraction was completed using the sonication technique, where approximately 1g of powdered leaves were immersed in 30 mL of extraction solvent and allowed to sonicate for 30 min. After sonication, filtration was performed using Whatman filter paper No.1. The rotary evaporator was used at 40 ± 1 °C to remove the solvents from the filtrate and followed by freeze-drying to remove the remaining solvent. The resultant dried extracts were stored at −80 ± 1 °C freezer prior to further analysis [1,14]. The samples were analyzed in parallel by NMR and GC-MS as well as evaluation of AGI activity.

### 4.4. Assay of α-Glucosidase Inhibitory Activity

The protocol reported by Murugesu et al. [1] was followed with slight modification. Quercetin (2 mg in 1 mL of DMSO) and PNPG (6 mg in 2 0mL 50 mM phosphate buffer, pH 6.5) were used as positive control and substrate, respectively. Glycine was prepared by dissolving 15 g in 100 distilled water and adjusted the pH to 10 using sodium hydroxide. The total volume in each well was 250 μL. The sample mixture contained 100 μL of 30 mM phosphate buffer, 10 μL of the sample, and 15 μL of the enzyme (0.02 U/μL). It was incubated for 5 min at room temperature before adding 75 μL of the substrate. The reaction was stopped after 15 min of incubation period at room temperature (24 ± 1 °C) by adding 50 μL of glycine. Similarly, the positive and negative controls were prepared by the addition of 10 μL of quercetin and DMSO to the mixture, respectively, instead of the sample. On the other hand, the blank mixture was prepared by the addition of 115 μL of 30 mM buffer without enzyme. After the addition of glycine, the absorbance was taken at 405 nm using a microplate reader (Tecan Nanoquant M 200, Grodig, Austria), and the percentage of inhibition was calculated by using the formula given below:Percentage of *α*-glucosidase inhibition = [(Absorbance of control−Absorbance of sample)/Absorbance of control] × 100(1)

IC_50_ values represented the plant extract concentration causing 50% *α*-glucosidase activity. It was calculated through regression analysis.

### 4.5. Derivatization Procedure for GC-MS

The sample was derivatized following the method described by Murugesu et al. [1] with slight modifications. As much as 25 mg of the sample was loaded to 2 mL centrifuge tube, and dissolved in 50 μL of pyridine, and vortexed for 5 min. Subsequently, 100 μL of methoxamine hydrochloride solution (20 mg in 1 mL of pyridine) was added to the mixture and then sonicated for 5 min. The sample mixture was then incubated at 60 °C for 2 h. A batch of 300 μL of MSTFA was added immediately, followed by further incubation at the same temperature for 30 min prior to overnight incubation at room temperature (27 ± 1 °C). The mixture was then filtered through a 0.45 μm syringe filter, and then finally transferred to the glass insert vial for injection to GCMS.

### 4.6. GC-MS Analysis

GC-MS analysis was carried out following the method described by Murugesu et al. [1] with some modifications. The GC-MS system was consisted of GC-MS (Agilent 6890) along with HP 5973 selective mass detector (Agilent). DB-5MS capillary column was used for the sample separation. The thickness, diameter, and length of the column were 0.25 μm, 250 μm, and 30.0 m, respectively. The pressure of the column was 11.41 psi. Two microliters of the derivatized sample mixture were injected into the system in splitless mode. Initially, the oven temperature was set at 85 °C. The initial temperature was gradually increased to the target temperature of 315 °C at a temperature increment rate of 2 °C per minute and then held for 5 min. The solvent delay was set to 6 min. Helium was used as a carrier gas, and the flow rate was set to 0.8 mL per min. The temperatures of the injector and ion source were set to 250 °C and 200 °C, respectively. Mass spectra were acquired using a full scan by setting parameters ranging from 50 to 550 *m*/*z*.

The raw data obtained from GC-MS was converted into a cdf.net format using ACD/Spec Manager v. 12.00 (Advanced Chemistry Development, Inc., ACD/Labs Ontario, Toronto, ON, Canada). The data were pre-processed for baseline correction, deconvolution, retention time correction, automatic peak detection and alignment utilizing Mzmine (VTT Technical Research Centre, Espoo, Finland) according to the following parameters: asymmetric baseline corrector (smoothing = 100,000, asymmetry = 0.5), centroid mass detection (noise level = 200), chromatogram building (minimum time span = 0.2 min, minimum height = 200, *m*/*z* tolerance = 1.0 mz or 2500 ppm), peak detection with filter width = 11, isotopic peak grouping (retention time tolerance = 0.2 min, monotonic shape, maximum charge = 1), duplicate peak filtering, linear normalization based on peak area, alignment (join aligner, weight for *m*/*z* = 85, weight for RT = 15, gap-filling (same RT and *m*/*z* range gap filler). Finally, the data were converted into an Excel file for further multivariate data analysis. The numbers of X-variables (preprocessed GC-MS signals) and the Y-variable (AGI activity) applied for multivariate data analysis were 1640 and 1, respectively. Through multivariate data analysis (partial least square) with the pareto scaling method, the AGI activity of twenty extracts were correlated with their respective spectral data that was obtained from GC-MS analysis. The X-variables located near to the AGI activity in the loading plot were considered to have AGI activity, under condition that it fulfill the validation criteria [13].

### 4.7. Bioactivity Confirmation of Pure Compounds

Owing to the limitation on the supply of the identified bioactive compounds, not all compounds could be checked for bioactivity confirmation. As a result, three pure compounds, namely palmitic acid, 1-monopalmitin, and *α*-tocopherol were purchased to assess the inhibition activity against *α*-glucosidase enzyme. IC_50_ of palmitic acid, methanol extract and quercetin were determined by following the method mentioned in Section 4.4. Instead of determining the IC_50_ of 1-monpalmitin and *α*-tocopherol, synergistic activity was examined by calculating the percentage of inhibition (Equation (1)). Five concentrations (4, 2, 1, 0.5, and 0.25 μg/mL) of 1-monopalmitin and *α*-tocopherol were used to investigate the percentage of inhibition against AG. To investigate the synergistic effect, the concentration of 2 μg/mL of methanol extract was used. Three groups were considered for this test. Group-1 consisted of a mixture of methanol extract and pure compounds, whereas the pure compounds and methanol extract individually were considered as Group 2 and Group 3, respectively.

### 4.8. H-NMR Sample Preparation

The sample for ^1^H-NMR analysis was prepared by following the method described by Mediani et al. [20] with slight modifications. A 10 mg of plant extract was accurately measured in 2 mL of Eppendorf tube, followed by adding 1:1 mixture of CH_3_OH-*d*_4_ (0.75 mL) and D_2_O (0.75 mL) containing 0.1% of trimethylsilylpropanoic acid (TSP). TSP was used as a reference standard. The sample–solvent mixture was then allowed to sonicate for 10 min, followed by centrifugation for 15 min at 13,000 rpm. Approximately 0.7 mL of the supernatant liquid was transferred into the 5 mm NMR tube (Norell, Sigma-Andrich, Oakville, ON, Canada) for further NMR analysis.

### 4.9. H-NMR Data Acquisition and Data Processing

The method described by Mediani et al. [20] was applied for ^1^H-NMR data acquisition and data processing with some modifications. In total, 20 samples were analyzed using Varian INOVA 500 megahertz NMR spectrometer (Varian Inc., Schiller Park, IL, USA). The ^1^H-NMR spectra were evaluated at 26 °C, and the 2D NMR included *J*-resolved for structural elucidation of the compounds present in the plant extracts. For each spectrum, 128 scans were used, required time was 14 min, relaxation delay 2.0 s, spectral width of 15 ppm and pulse width 21.0 microsecond (90°) were considered. Deuterium oxide and 0.1% TSP were used as internal lock and standard (δ 0.00 ppm) for calibration, respectively. The phase and baseline corrections were performed using Chenomix software (Chenomx NMR Suite 5.1 Professional, Edmonton, AB, Canada). The chemical shift region of methanol (δ 3.27–δ 3.33) and water (δ 4.68–δ 4.88) were excluded from the raw data. The raw data were binned to ASCII files with the spectral width at δ 0.04 ppm and a region of δ 0.5 ppm to δ 10 ppm. After binning the NMR spectra, multivariate data analysis was carried out using SIMCA P + 14.0 (Umetrics, Umeå, Sweden) software, where the pareto scaling method was applied. The numbers of X-variables (binned NMR signals) and Y-variable used (AGI activity) for multivariate data analysis were 232 and 1, respectively. The NMR spectral data of all plant extracts were correlated with the AGI activity of the respective extracts via multivariate data analysis (orthogonal partial least square) using SIMCA P + 14.0 (Umetrics, Umeå, Sweden) software.

### 4.10. Metabolite Assignment

The metabolite assignment was performed based on the GC-MS and NMR spectra. The fragment *m*/*z* spectra of each compound obtained from the GC-MS analysis were compared to the National Institute of Standards and Technology (NIST) 2014 database.

The identification of the metabolites was also performed based on the comparison of the characteristic signals found in ^1^H-NMR spectra of the *P. malayana* leaves extract with the reported literature, Chenomix database (Chenomx NMR Suite 5.1 Professional, Edmonton, AB, Canada), and human metabolome database (www.hmdb.ca; accessed on 5 March 2021). The 2D *J*-resolved was also used to support the identification of the metabolites.

### 4.11. In-Silico Study

AutoDock Vina (version 1.1.2) has been used to predict the binding interactions and binding affinity between the determined metabolites and the *α*-glucosidase enzyme. The crystal structure of the enzyme was collected from Protein Data Bank (PDB) (PDB code: 3A4A) [65,72,75,76,77,78]. The three-dimensional (3D) structures of 1,3,5-benzenetriol, 1-monopalmitin were collected from the National Institute of Science and Technology database, and 24-epicampesterol, *α*-tocopherol, *β*-tocopherol, cholesta-1,3-diene, palmitic acid, and quercetin were obtained from Pub Chem database as .sdf format. The structure in .sdf format was converted into *.pdb format using Open Babel (version 2.3.1) [65]. AutoDock Tools (version 1.5.6) was used to add Gasteiger charge and saved in .pdbqt format prior to molecular docking [75]. In addition, the rotatable bonds have been fixed by AutoDock Tools. The co-crystallized ligand, *α*-D-glucose (ADG) has been re-docked into the enzyme (3A4A). Root mean square deviation (RMSD) values were used to select the top-leveled docking conformations by comparing them with the actual crystallographic conformation. For the docking of positive control, quercetin, and the other identified compounds against the enzyme, similar parameters as used in the control docking were used [37,79]. The coordinates of the grid box were positioned at 21.272, −0.751, and 18.634. The grid box dimension was 20 Å, 26 Å, and 22 Å for X, Y, and Z, respectively, and the level of exhaustiveness was set as 16. The docking was conducted three times. The enzyme crystal structure was protonated at a pH of 6.5 to mimic the actual condition of the bioassay using the PDB2PQR server version 2.0.0. [65,80]. PyMOL TM 1.7.4.5 (Schrödinger, LLC., New York, NY, USA) was used to render the 3D superimposed diagram of the control ligand, quercetin, and the detected compounds [65]. The molecular interactions were investigated using Biovia Discovery Studio Visualizer (San Diego, CA, USA) [37,65,79].

### 4.12. Statistical Analysis

Minitab 16 (Minitab Inc., State College, PA, USA) has been used for statistical analysis in order to evaluate the difference of AGI activity among the extracts and selected putative compounds. One-way variances (ANOVA) and Tukey’s test were used to analyze the data with a confidence interval of 95% and considered significant at *p* < 0.05. The data are interpreted as the mean ± standard deviation (SD) with *n* = 4. Values represented with different superscripts are significantly different.

Multivariate data analysis was carried out using SIMCA P + 14.0 (Umetrics, Umeå, Sweden) software in order to discriminate the samples based on their GC-MS/NMR signal profiles as well as the AGI activity. This analysis was also used to reveal the GC-MS/NMR signals correlation to the AGI activity. The pareto scaling method was applied and fitted with PLS and OPLS technique. The validation parameters such as jack-knifing error bar, variable influence on projections, and R^2^ value of observed vs. predicted AGI activity were applied to check the validity of the models.

## 5. Conclusions

The present study identified ten tentative bioactive metabolites from the methanolic extracts of *P. malayana* leaves by GC-MS based and NMR-based metabolomics, where some of them are for the first time reported in this plant. The methanolic extract which contains the detected metabolites, namely 1,3,5-benzenetriol, palmitic acid, cholesta-7,9(11)-diene-3-ol, 1-monopalmitin, *β*-tocopherol, α-tocopherol, 24-epicampesterol, stigmast-5-ene, 4-hydroxyphenylpyruvic acid, and glutamine, exhibited strong AGI activity. The possible binding modes of these identified compounds to inhibit the enzyme were elucidated through molecular docking study. This finding scientifically supports the traditional claim of the usage of this plant as a medicinal plant with anti-diabetic properties. The unknown AGI inhibitors revealed in this study could be purified in the future in order to reveal their chemical structures.

## Figures and Tables

**Figure 1 pharmaceuticals-14-00978-f001:**
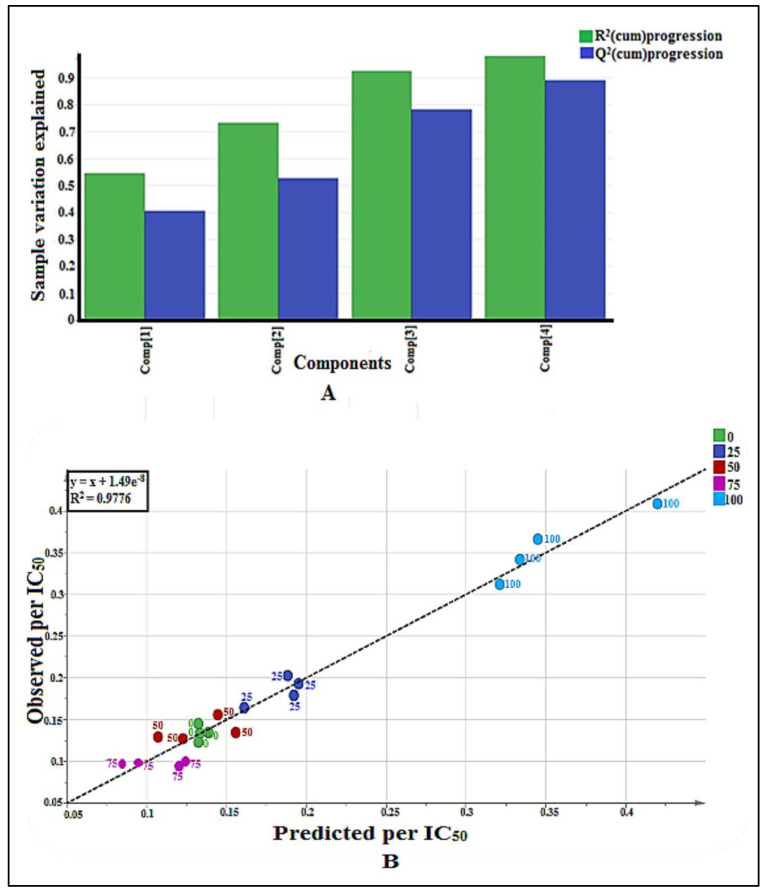
(**A**) Summary of fit of established PLS model for *P. malayana* leaves extracts (*n* = 4), where four components involved and this plot indicates the fitness of the model. (**B**) Observed vs. predicted AGI activity with R^2^ value 0.9776 from 20 extracts of *P. malayana* leaves (*n* = 4), indicates the model is valid.

**Figure 2 pharmaceuticals-14-00978-f002:**
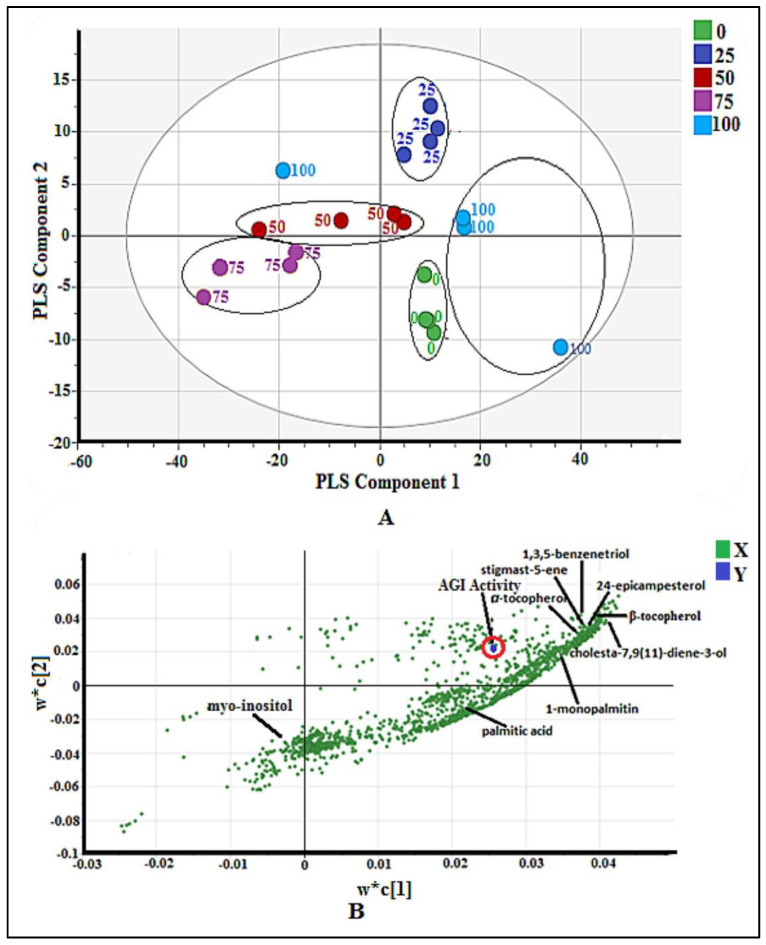
(**A**) Score scatter plot of validated PLS model for 20 extracts of *P. malayana* leaves (*n* = 4), indicates the clear separation among the five groups (0, 25, 50, 75, and 100% *v*/*v* methanol-water). (**B**) The loading scatter plot of PLS model, indicates the correlation between the AGI activity (Y variables) and the samples’ *m*/*z* value (X variables).

**Figure 3 pharmaceuticals-14-00978-f003:**
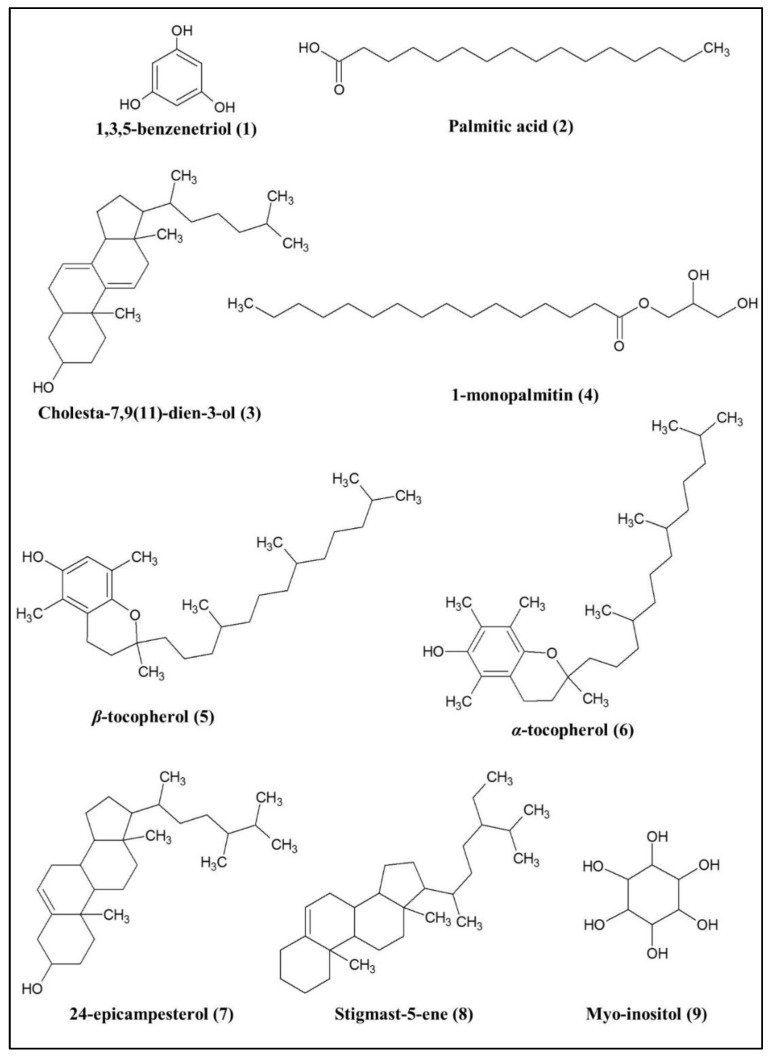
Chemical structures of putative metabolites identified by GC-MS-based metabolomics.

**Figure 4 pharmaceuticals-14-00978-f004:**
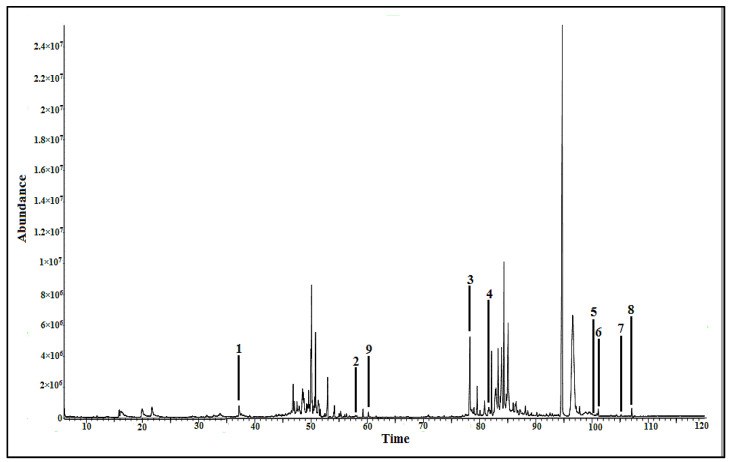
GC-MS chromatogram of the methanol extract of *P. malayana* leaves. Labelled peaks: **1**-1,3,5-benzenetriol, **2**-palmitic acid, **3**-cholesta-7,9(11)-diene-3-ol, **4**-1-monopalmitin, **5**-*β*-tocopherol, **6**-*α*-tocopherol, **7**-24-epicampesterol, **8**-stigmast-5-ene, and **9**-myo-inositol. The fragment *m*/*z* spectra of all compounds were compared to the NIST 14 database with similarity index above 90%.

**Figure 5 pharmaceuticals-14-00978-f005:**
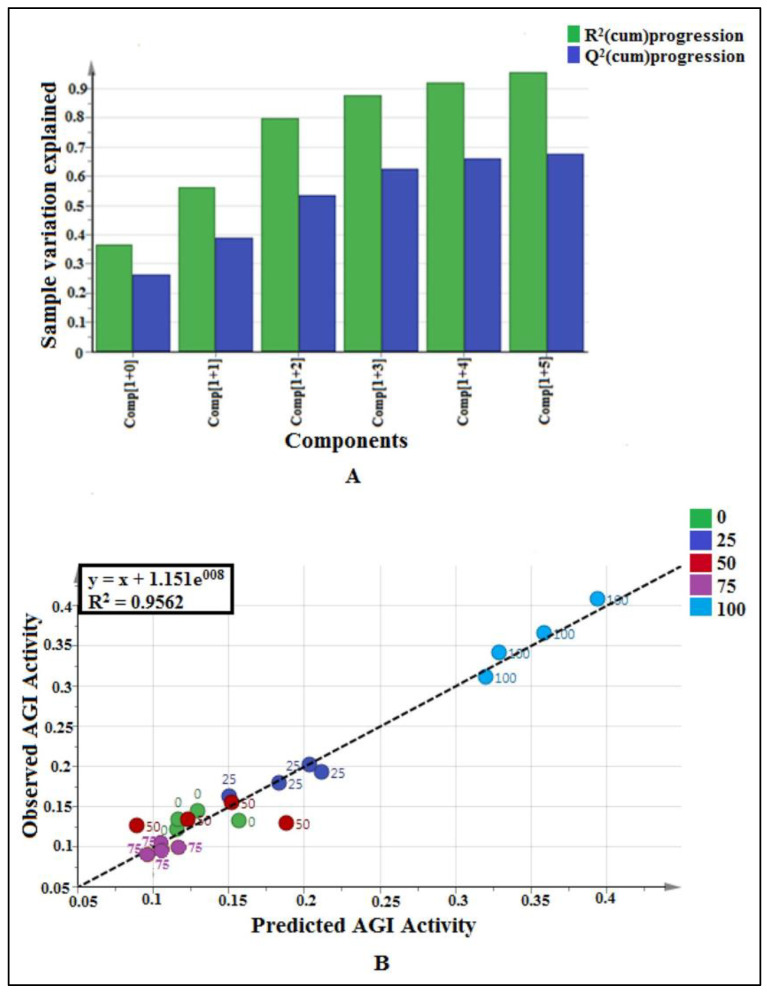
(**A**) Summary of fit of established OPLS model for *P. malayana* leaves extracts (*n* = 4), where six principle components involved and this plot indicate the fitness of the model. (**B**) Observed vs. predicted AGI activity with R^2^ value 0.9562 from 20 extracts of *P. malayana* leaves (*n* = 4), indicating the model was valid.

**Figure 6 pharmaceuticals-14-00978-f006:**
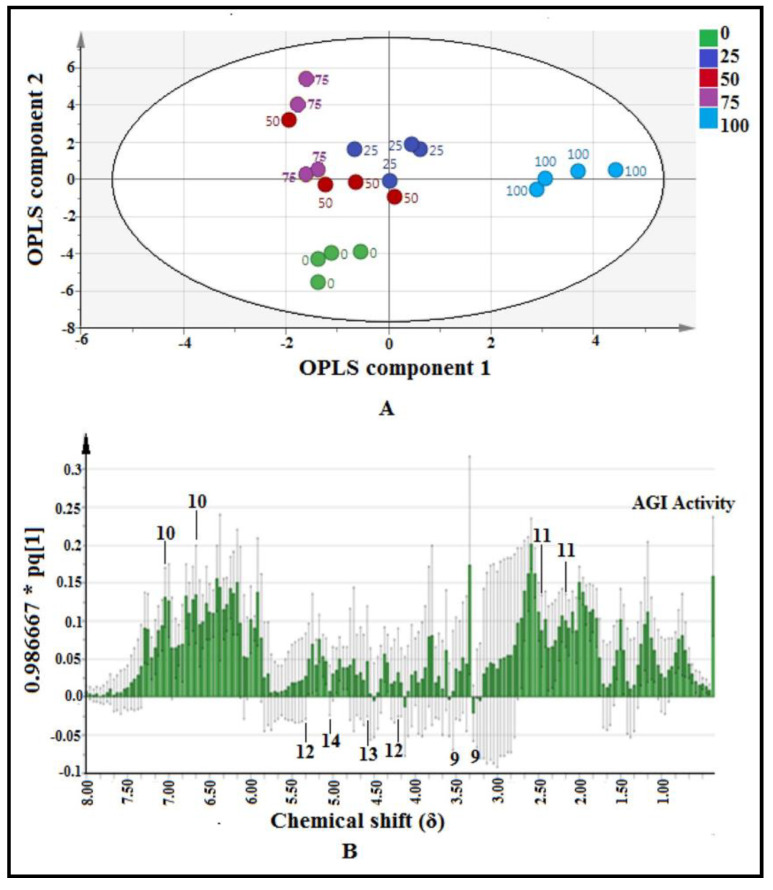
(**A**) Score scatter plot of validated OPLS model for 20 extracts of *P. malayana* leaves (*n* = 4), indicating the clear separation among the five groups (0, 25, 50, 75, and 100% *v*/*v* methanol-water). (**B**) The loading column plot of OPLS model, indicating the correlation between AGI activity (Y variables) and the samples’ *δ* value (X variables).

**Figure 7 pharmaceuticals-14-00978-f007:**
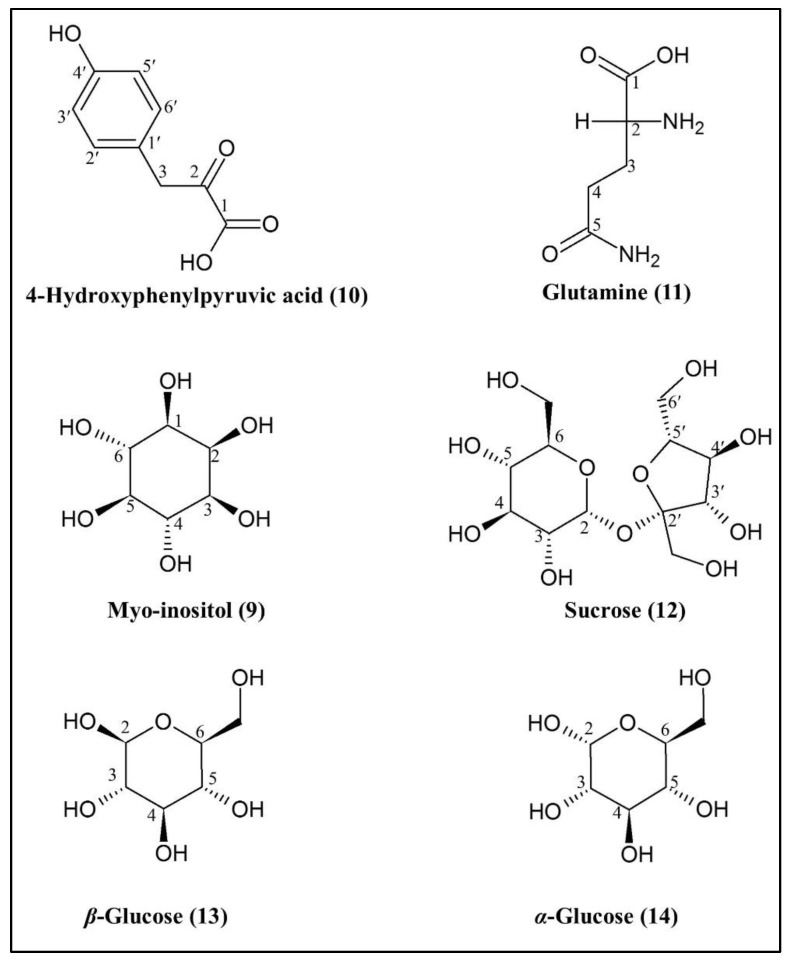
Chemical structures of putative metabolites identified by NMR-based metabolomics.

**Figure 8 pharmaceuticals-14-00978-f008:**
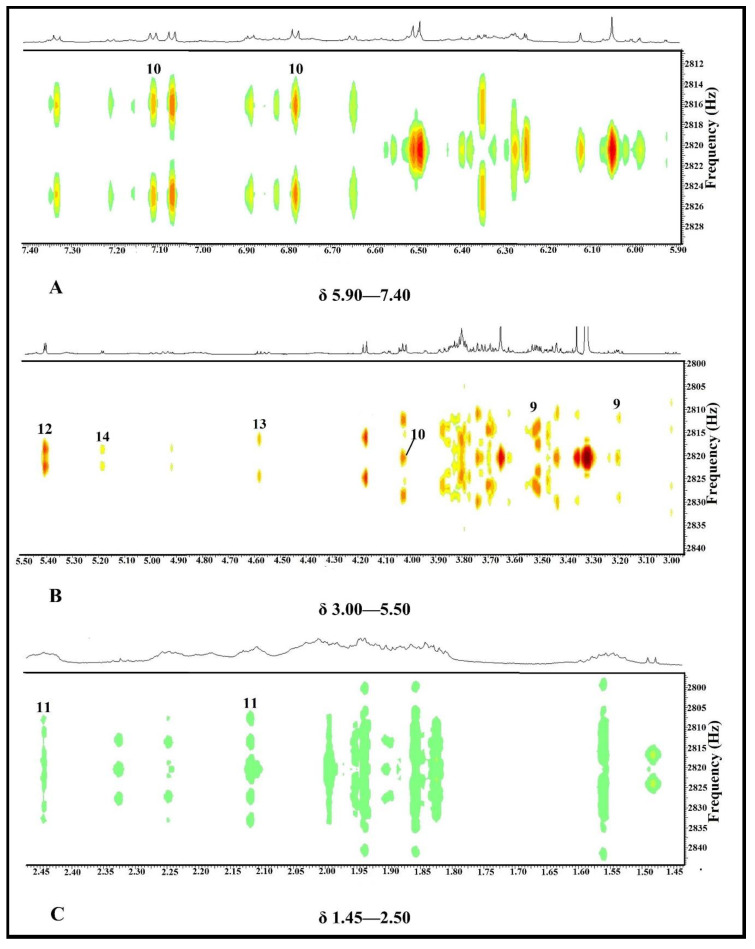
500 MHz 2D *J*-resolved NMR spectrum of *P. malayana* leaves extract based on which the multiplicity and coupling constants have been analyzed. (**A**) 2D *J*-resolved NMR spectrum between the range of δ 5.90–7.40. (**B**) 2D *J*-resolved NMR spectra between the range of δ 3.00–5.50. (**C**) 2D *J*-resolved NMR spectrum between the range of δ 1.45–2.50.

**Figure 9 pharmaceuticals-14-00978-f009:**
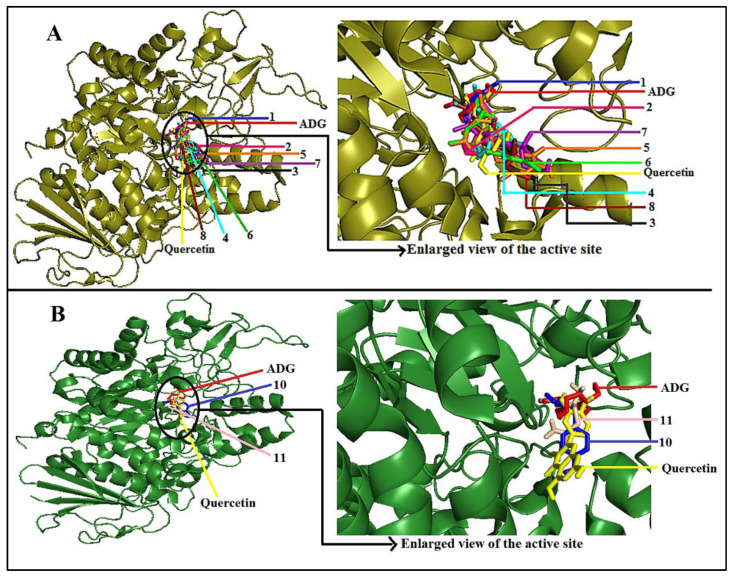
(**A**) 3D superimposed diagram of the bioactive compounds (**1**–**8**) identified by GC-MS analysis, ADG, and quercetin in enzyme (3A4A). (**B**) 3D superimposed diagram of the bioactive compounds (**10** and **11**) identified by NMR analysis, ADG, and quercetin in the enzyme (3A4A). Figure (**A**,**B**) indicate the simulated binding site of the putative bioactive compounds on the *α*-glucosidase enzyme (3A4A).

**Figure 10 pharmaceuticals-14-00978-f010:**
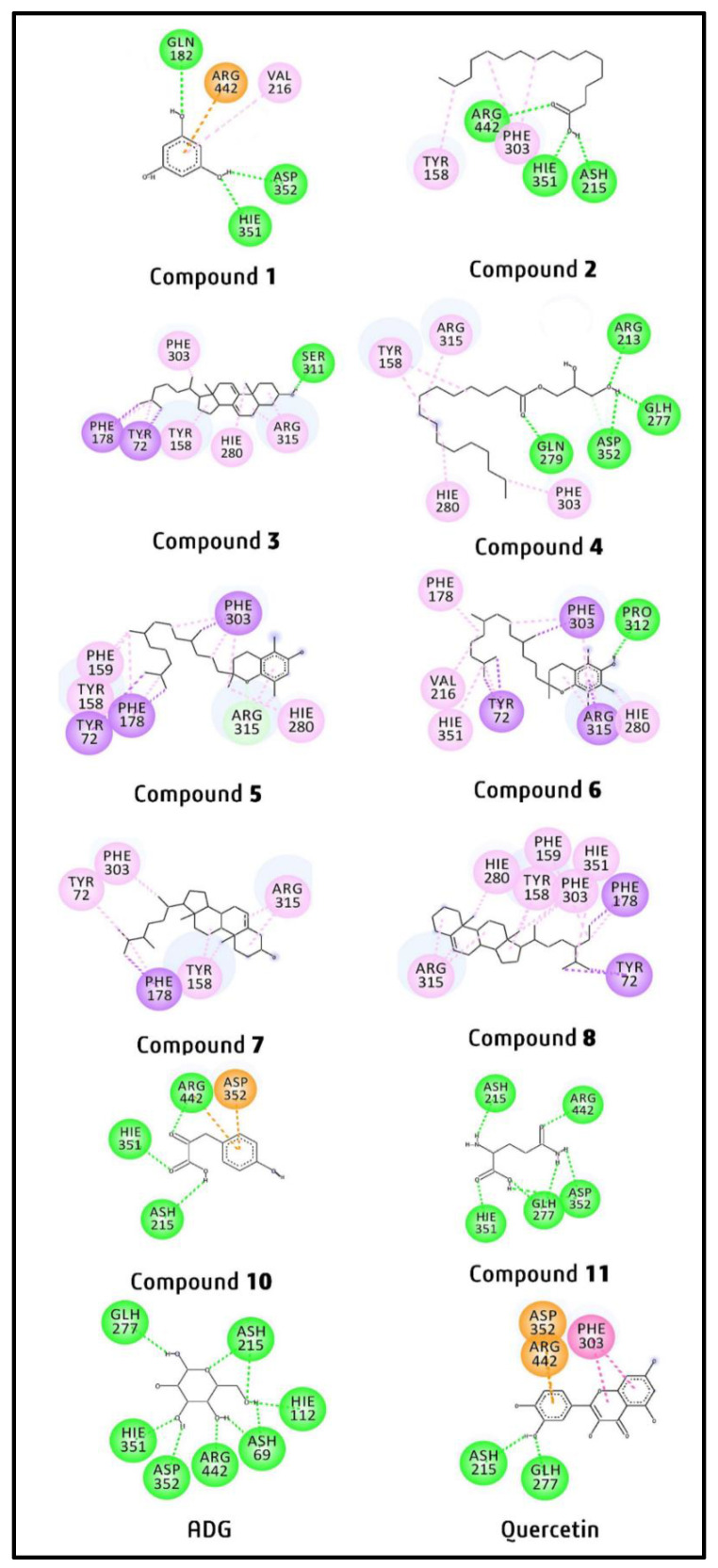
Predicted 2D binding interactions of the docked compounds **1**–**8** and **10**–**11** with *α*-glucosidase (3A4A).

**Table 1 pharmaceuticals-14-00978-t001:** The percentage of *α*-glucosidase inhibition of *P. malayana* leaves extracts at the concentration of 5 µg/mL.

Concentration of Methanol in Water	(%) of Inhibition *
0%	39.6 ± 3.32 ^c^
25%	49.6 ± 2.37 ^b^
50%	42.0 ± 4.04 ^b,c^
75%	36.1 ± 3.37 ^c^
100%	71.7 ± 6.56 ^a^
Quercetin	74.7 ± 1.14 ^a^

* The values represent mean + SD, *n* = 4. The values that do not share the same letter are significantly different with *p*-value < 0.05. Values represent with different superscripts are significantly different, as measured by Tukey’s comparison test.

**Table 2 pharmaceuticals-14-00978-t002:** Putative metabolites detected by GC-MS analysis.

Compound No.	Putative Metabolites	RT (min)	% of Area	SI	MF
**1**	1,3,5-benzenetriol	37.213	0.61	98	C_6_H_6_O_3_
**2**	Palmitic acid	58.142	0.04	98	C_16_H_32_O_2_
**3**	Cholesta-7,9(11)-diene-3-ol	78.294	3.42	94	C_28_H_48_O
**4**	1-monopalmitin	81.935	0.17	93	C_19_H_38_O_4_
**5**	*β*-tocopherol	100.870	0.10	93	C_28_H_48_O_2_
**6**	*α*-tocopherol	101.104	0.24	98	C_29_H_50_O_2_
**7**	24-epicampesterol	104.362	0.03	90	C_28_H_48_O
**8**	Stigmast-5-ene	107.083	0.31	99	C_29_H_50_
**9**	Myo-inositol	60.234	0.15	94	C_6_H_12_O_6_

RT = Retention time; SI = Similarity Index; and MF = Molecular formula.

**Table 3 pharmaceuticals-14-00978-t003:** IC_50_ of Palmitic acid.

Sample Name	IC_50_ (μg/mL) *
Palmitic acid	8.04 ± 0.12 ^a^
Methanol extracts	2.83 ± 0.32 ^b^
Quercetin	1.86 ± 0.04 ^c^

* The values represent mean + SD, *n* = 4. The values that do not share the same letter are significantly different with *p*-value < 0.05. Values represent with different superscripts are significantly different, as measured by Tukey’s comparison test.

**Table 4 pharmaceuticals-14-00978-t004:** Synergistic activity of 1-monopalmitin.

Concentrations of 1-Monopalmitin (μg/mL) Added to 2 μg/mL of Methanol Extract	% of AGI *	Concentrations of 1-Monopalmitin (μg/mL)	% of AGI
4	75.59 ± 1.95 ^a^	4	0
2	61.89 ± 0.77 ^b^	2	0
1	59.03 ± 1.23 ^c^	1	0
0.5	41.14 ± 1.18 ^d^	0.5	0
0.25	34.36 ± 0.49 ^e^	0.25	0
0	31.84 ± 0.87 ^e^	-	-

* The values represent mean + SD, *n* = 3. The values that do not share the same letter are significantly different with *p*-value < 0.05. Values represent with different superscripts are significantly different, as measured by Tukey’s comparison test.

**Table 5 pharmaceuticals-14-00978-t005:** Synergistic activity of *α*-tocopherol.

Concentrations of *α*-Tocopherol (μg/mL) Added to 2 μg/mL of Methanol Extract	% of AGI *	Concentrations of *α*-Tocopherol (μg/mL)	% of AGI *
4	97.08 ± 0.60 ^a^	4	10.93 ± 0.63 ^g^
2	90.94 ± 0.91 ^b^	2	7.84 ± 0.82 ^h^
1	85.22 ± 0.49 ^c^	1	6.07 ± 0.85 ^h^
0.5	74.55 ± 0.37 ^d^	0.5	3.61 ± 0.85 ^i^
0.25	56.99 ± 1.46 ^e^	0.25	0
0	31.18 ± 1.09 ^f^	-	-

* The values represent mean + SD, *n* = 3. The values that do not share the same letter are significantly different with *p*-value < 0.05. Values represent with different superscripts are significantly different, as measured by Tukey’s comparison test.

**Table 6 pharmaceuticals-14-00978-t006:** The ^1^H-NMR chemical shifts, proton number, multiplicity, and coupling constant of the identified compounds in *Psychotria malayana* extract.

Compound	Putative Metabolites	Proton Number	Chemical Shift (ppm)	Multiplicity *	Coupling Constant (*J*)	References
**9**	Myo-inositol	H-1 H-5	3.51 3.20	dd t	10.0 Hz; 4.0 Hz 9.0 Hz	[25,26,27]
**10**	4-hydroxyphenyl-pyruvic acid	H-3′; H-5′ H-2′; H-6′ H-3	6.78 7.11 4.03	d d s	8.0 Hz 8.0 Hz -	[28,29]
**11**	Glutamine	H-4 H-3	2.44 2.12	m m	- -	[26,30,31,32,33,34]
**12**	Sucrose	H-2 (glucose moiety)	5.40	d	4.0 Hz	[26,32,35,36]
**13**	*β*-glucose	H-2	4.58	d	8.0 Hz	[26,32,35,36]
**14**	*α*-glucose	H-2	5.18	d	4.0 Hz	[26,32,35,36]

* s: singlet; d: doublet; t: triplet; dd: doublet of doublet; and m: multiplet.

**Table 7 pharmaceuticals-14-00978-t007:** Binding affinity values of *α*-glucosidase enzyme (3A4A) with ADG, quercetin and the identified ten active compounds.

Compound	Binding Affinity (kcal/mol)
Control ligand (ADG)	−6.0
Quercetin	−8.4
**1**	−5.5
**2**	−6.1
**3**	−9.1
**4**	−6.1
**5**	−8.6
**6**	−7.9
**7**	−7.7
**8**	−9.4
**10**	−6.5
**11**	−5.8

## Data Availability

Data is contained within the article.

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
