# Peer review of "GC-MS- and NMR-Based Metabolomics and Molecular Docking Reveal the Potential Alpha-Glucosidase Inhibitors from Psychotria malayana Jack Leaves"

_pharmaceuticals, 2021, doi:10.3390/ph14100978_

Round 1

Reviewer 1 Report

Overall: The authors present an interesting and comprehensive study by identifying metabolites in a medicinal plant and connecting the metabolite first through statistical and enzyme activity approaches and then with molecular docking. My suggestions for improvement are listed by section. 

Introduction (section 1):

  • Biostats employed to screen for activity should be mentioned including how the authors and others in the field are getting activity from metabolomics (not all metabolites are 'bioactive').
  • The authors should also define bioactive here (they mean druggable in humans but other definitions exist such as activity in microbes)  
  • Minor grammar like line 60: 'is accounting for', should be 'accounts for' and other similar small mistakes throughout the manuscript should be corrected. 

Methods (section 4):

  • Replicates? Biological and technical? Power analysis? How many plants? How many leaves? How do they know they are sufficiently powered to perform such analyses like multivariate? 2 tables or figures mention n=4 but it is not described if those are technical replicates or exactly what n represents and not described for most assays. 
    • For untargeted metabolomics, n=10 is preferred but at least n=5. 
  • Line 723: Which samples are derivatized for GC-MS? Each of the different methanol extracts? How many replicates? How were they normalized? Protein concentration? 
  • Line 744-6: Just the chromatograms, not the fragment masses were compared to NIST? It is odd to derivatize and perform GC-MS and not present nor compare mass data. This really only reflects GC data. Including the MS portion would provide a lot more certainty on the metabolite identification. 
  • Line 761:  The bioactive compound identification procedure is not described (other than loosely mentioning NIST). How were metabolites identified? How were those metabolites labeled as bioactive? 
  • Line 766; Why was synergistic testing performed? What is the rationale for a combined molecule inhibition compared to individual? 
  • Line 687- St. Louis is spelled wrong. Line 834 is missing a period and other similar small typos throughout the manuscript should be corrected. 
  • Line 774: Which samples are being analyzed by NMR? Eluate from GC-MS? Separate preparations? 
  • Section 4.10: how are the metabolites from NMR related to the metabolites from GC-MS? Why weren't GC-MS values also used for identification?   
  • section 2.12 (line 828- should be 4.12?) providing a rationale for the tests and what samples and what questions are being asked with each test would enhance this section. 
  • Data Availability statement? 

Results (section 2)

  • Since the results are presented before the methods, each subsection should start with a short summary of what test was performed and the rationale for it. 
    • This is especially true for the statistical sections as the methods section was very short and the captions are also too short. What are you testing with these computational approaches? Why and how? What samples are you testing (like Line 127 and other places)? What data was fed into these models? What do the results mean for your samples? The results are not connected back to anything about leaves or bioactive metabolites or enzymes in diabetes. 
    • Figure 2 needs much more description and biological interpretation. Why were these tests run? What data were they run on? What do they mean? How are metabolites identified in Figure 2? Identifying molecules has not been mentioned in results yet. 
    • Lines 175-180 (and others) annotated does not equal identification in metabolomics. also how know bioactive? 
    • Where is the reference data for GC-MS (from NIST). Score is nice but reference data is better. 
  • Line 192: How were the IC50 values obtained (enzyme activity not mentioned in results). How were metabolites selected for inclusion in Table 3 from Table 2/Figure 3?  
  • Again 243-245: What samples/ Why this stats method? What question is being asked? How is the NMR data related to the GC-MS data? 
  • Section 2.3.2- why where these compounds chosen for identification? How know bioactive? Were IC50's measured for these? The rationale for all steps should be described more throughout results. 
  • Figure captions: all need to be much more descriptive. Figure captions should be a stand-alone description of the figure.  Units in some plots are missing and incomplete axes labels. n=? is missing for most.  

Discussion (section 3)

  • The discussion would benefit from subsection headings to convey main points.  

Author Response

We agree with the general thrust of the reviewer’ comments – that there were many opportunities to improve the originally submitted manuscript – and have spent considerable time rewriting our original draft to address their concerns. As demonstrated by the responses below, we have carefully considered the reviewer’ points and addressed them by extensively rewriting the manuscript to improve its flow, coherence, and scientific content. All revised parts were marked with a yellow background in the manuscript. We are confident that the revised and restructured manuscript does not suffer from the problems identified by the reviewer in the original draft, and hope you will find it worthy for publication.

Comments of Reviewer 1

Responses

Overall: The authors present an interesting and comprehensive study by identifying metabolites in a medicinal plant and connecting the metabolite first through statistical and enzyme activity approaches and then with molecular docking. My suggestions for improvement are listed by section. 

Introduction (section 1):

Biostats employed to screen for activity should be mentioned including how the authors and others in the field are getting activity from metabolomics (not all metabolites are 'bioactive').

Amendment has been made as per suggestion. We have inserted a sentence explaining the way how multivariate data analysis pinpoints the bioactive compounds (see section 1: line 95-99).

The authors should also define bioactive here (they mean druggable in humans but other definitions exist such as activity in microbes)

Amendment has been made as per suggestion (see section 1: line 89-90).

Minor grammar like line 60: 'is accounting for', should be 'accounts for' and other similar small mistakes throughout the manuscript should be corrected. 

Amendment has been made as per suggestion (see section 1: line 65).

Methods (section 4):

Replicates? Biological and technical? Power analysis? How many plants? How many leaves? How do they know they are sufficiently powered to perform such analyses like multivariate? 2 tables or figures mention n=4 but it is not described if those are technical replicates or exactly what n represents and not described for most assays. 

For untargeted metabolomics, n=10 is preferred but at least n=5.

The number of X-variables used for multivariate data analysis of NMR and GC-MS data were 232 and 1640, respectively.

While, the number of Y-variable was 1 (see section 4.6: line 782-788; section 4.9: line 827-831) This number of variables is sufficient in performing the multivariate data analysis, which was confirmed by the validation parameters such as jack-knifing error bar, variable influence on projections, and R2 value of observed vs predicted AGI activity (see section 4.12: line 876-878),

Although the technical replicate in this study was only four (n=4), the above validation parameters confirmed its validity.

Line 723: Which samples are derivatized for GC-MS? Each of the different methanol extracts?

How many replicates?

How were they normalized? Protein concentration?

All different methanol extracts were derivatized for GC-MS analysis with four replicates. Amendment has been made as per suggestion in section 4.3: line 721.

The GC-MS raw data were pre-processed for baseline correction, deconvolution, retention time correction, automatic peak detection and alignment utilizing Mzmine (VTT Technical Research Centre, Finland) according  to the following parameters: asymmetric baseline corrector (smoothing=100,000, asymmetry= 0.5), centroid mass detection (noise level=200), chromatogram building (minimum time span = 0.2 min, minimum height = 200, m/z tolerance= 1.0 mz or 2500 ppm), peak detection with filter width= 11, isotopic peak grouping (retention time tolerance= 0.2 min, monotonic shape, maximum charge= 1), duplicate peak filtering, linear normalization based on peak area, alignment (join aligner, weight for m/z = 85, weight for RT= 15, gap-filling (same RT and m/z range gap filler) (see section:4.6: line 770-781).

While the protein concentration was not considered during GC-MS analysis.

Line 744-6: Just the chromatograms, not the fragment masses were compared to NIST?

It is odd to derivatize and perform GC-MS and not present nor compare mass data. This really only reflects GC data. Including the MS portion would provide a lot more certainty on the metabolite identification.

Amendment has been made as per suggestion. The identification of compounds was done through comparing of mass fragments to NIST database (see section 4.10: line 834-835). The fragment m/z spectra of each putative compound are shown in Figure S3 – S10.

Line 761:  The bioactive compound identification procedure is not described (other than loosely mentioning NIST). How were metabolites identified?

The description of the identification procedure has been mentioned in section 4.6: line 782-788 and section 4.10: line 834-835.

How were those metabolites labeled as bioactive? 

The identification of bioactive compounds employed a supervised multivariate data analysis (partial least square) in correlating X-variables (instrumental signals from the metabolites) to Y-variables (bioactivity of the samples). The X-variables situated close to the Y-variables in the loading plot, indicating the positive correlation to the bioactivity (see section 1: line 95-99; section 2.2.1: line 188-191; and section 4.6: line 786-788.

Line 766; Why was synergistic testing performed?

What is the rationale for a combined molecule inhibition compared to individual?

The reason for testing synergistic activity is mentioned in section 2.2.3: line 236-238 and section 3: line 528-532.

The rationale has been mentioned in section 3: line 526-541.

Line 687- St. Louis is spelled wrong.

Amendment has been made as per suggestion (see section 4.1: line 709).

Line 834 is missing a period and other similar small typos throughout the manuscript should be corrected. 

Amendment has been made as per suggestion (see section 4.12: line 876).

Line 774: Which samples are being analyzed by NMR? Eluate from GC-MS? Separate preparations?

The same samples were analyzed in parallel by NMR and GC-MS as mentioned in section 4.3: line 727-728.

Section 4.10: how are the metabolites from NMR related to the metabolites from GC-MS? Why weren't GC-MS values also used for identification?

We have added two sentences related to the GC-MS analysis in section 4.10: line 833-835.

section 2.12 (line 828- should be 4.12?)

Amendment has been made as per suggestion (see section 4.12: line 865).

providing a rationale for the tests and what samples and what questions are being asked with each test would enhance this section. 

Data Availability statement? 

Amendment has been made as per suggestion (see section 4.12: line 866-868, line 870-871, line 873-875, and line 876-878).

Results (section 2)

Since the results are presented before the methods, each subsection should start with a short summary of what test was performed and the rationale for it.

Amendment has been made accordingly (see section 2: line 118-124).

This is especially true for the statistical sections as the methods section was very short and the captions are also too short.

We have improved our statistical section under methodology part accordingly (see section 4.12: line 866-868, line 870-871, line 873-875, and line 876-878)

What are you testing with these computational approaches? Why and how?

These computational approaches aimed to reveal the bioactive compounds possessing AGI activity from the extracts (see section1: line 89-99; see section 2: line 118-124; section 2.2.1: line 145-146 and 188-195; and section 2.3.1: line 269-270, line 313-325). The supervised multivariate data analysis was used to fulfill this goal (see section 1: line 95-99).

What samples are you testing (like Line 127 and other places)?

We have changed the word “sample” to “plant extract” (see section 2.2.1: line 146).

What data was fed into these models?

GC-MS data (X variables) and AGI activity (Y variable) of each sample were fed into this PLS model to analyze the correlation (see section 2.2.1: line 145-146 and section 4.6: line 782-783).

What do the results mean for your samples?

The results are not connected back to anything about leaves or bioactive metabolites or enzymes in diabetes

The result showed the connection between the bioactive compounds present in the plant extracts and the AGI activity (see section 2.2.1: line188-191; section 2.3.1: line 313-315). The connection among the results is briefly described in section 2: line 118-124.

Figure 2 needs much more description and biological interpretation. Why were these tests run? What data were they run on? What do they mean? How are metabolites identified in Figure 2? Identifying molecules has not been mentioned in results yet.

Figure 2A shows the clustering among the plant extracts (see section 2.2.1: line 183-186), while figure 2B pin points the GC-MS signals (X-variable) corresponding to the AGI activity (Y-variable) (see section 2.2.1: line188-191). The X-variables close to the AGI activity in the loading plot (figure 2B) were considered as the GC-MS signals positively correlating to the AGI activity (see section 2.2.1: line 190-191). While the identification of the putative compounds was performed through the comparison between the fragment m/z spectra and the NIST database (see section 2.2.1: line 194-195).

Lines 175-180 (and others) annotated does not equal identification in metabolomics. also how know bioactive?

We have changed the term “annotated” to “identified” accordingly (see section 2.2.2: line 199).

The bioactivity of the compounds was predicted through multivariate data analysis by means of correlation between the signal profile from the compounds and the AGI activity (see section 2.2.1: line 188-191).

Where is the reference data for GC-MS (from NIST). Score is nice but reference data is better.

Amendment has been made as per suggestion. The fragment m/z spectra of each putative compound are shown in Figure S3 – S10.

Line 192: How were the IC50 values obtained (enzyme activity not mentioned in results).

The method of IC50 values has been described in section 4.4: line 745-746; section 4.7: line 793-794. While table 3 shows the results.

How were metabolites selected for inclusion in Table 3 from Table 2/Figure 3? 

The selection for inclusion criteria has been mentioned in section 4.7: line 790-793.

Again 243-245: What samples/ Why this stats method? What question is being asked?

We have changed the word “samples” to “plant extracts” (see section 2.3.1: line 270).

This approach employs a supervised multivariate data analysis (orthogonal partial least square) correlating X-variables (NMR signals from the metabolites) to Y-variables (AGI activity of the plant extracts). The X-variables situated close to the Y-variables in the loading column plot, indicating the positive correlation to the bioactivity (see section 1: line 95-99 and section 2.3.1: line 313-315). This step is designed to reveal the bioactive compounds possessing AGI activity detected by NMR.

How is the NMR data related to the GC-MS data? 

One compound (myo-inositol) was detected by both GC-MS and NMR, while the others can only be detected by one analytical instrument. This is due to the nature of each instrument. NMR is not sensitive as GC-MS, while GC-MS can only detected the volatile or derivatised compound (see section 2.3.2: line 336-337).

Section 2.3.2- why where these compounds chosen for identification?

The NMR signals of these compounds fit to the reported literature, Chenomix database (Chenomx NMR Suite 5.1 Professional, Ed-833 monton, Canada), and human metabolome database (www.hmdb.ca) (see section 4.10: line 836-840).

How know bioactive?

Several steps were followed to identify the bioactive compounds:

-        The NMR signals from the compounds close to the AGI activity in the loading plot were statistically considered having the AGI activity (see section 2.3.1: and line 313-315).

-        The NMR profiles of the above selected signals (chemical shifts, multiplicity, coupling constant, and intensity) were then compared to the available database in order to identify the chemical structures (see section 2.3.2: line 331-348).

-        The in-silico docking study predicted the AGI activity of the above selected compounds (see section 2.4: line 357-354).

Were IC50's measured for these?

The IC50 were not measured for the compounds identified by NMR in this study due to the absent of the commercial supply of the identified compounds. However, the in-silico docking study can predict the AGI activity of these compounds.

The rationale for all steps should be described more throughout results

Amendment has been made accordingly (see section 2: line 118-124).

Figure captions: all need to be much more descriptive. Figure captions should be a stand-alone description of the figure.  Units in some plots are missing and incomplete axes labels. n=? is missing for most.

Amendment has been made as per suggestion from Figure 1 – Figure 10.

Discussion (section 3)

The discussion would benefit from subsection headings to convey main points. 

The subsections heading have been created accordingly (see section 3.1-3.3).

Reviewer 2 Report

Tanzina Sharmin Nipun and co-authors presented a study about alpha-glucosidase inhibitors from Psychotria malayana jack leaves identifying ten putative bioactive compounds in this plant for the first time. In my opinion, the study itself is interesting for scientific community and the readers of Pharmaceuticals and could be published in this journal. Below you can find some of my corrections/ comments / suggestion to improve this manuscript:

  • Page 6: Concerning the identification of bioactive compounds: If you compare your chromatogram in Figure 4 to chromatogram of the pure compounds 1-9 (for example purchased standards) using the same methods, this would definitely confirm the presence of tested compounds in malayana leaves extracts if retention times match. Please explain more in details how did you annotate the bioactive compounds – based on what (retention times using the same method as in National Institute of Standards database?)?
  • Page 13, 2.3.2. Identification of Putative Bioactive and Other Metabolites, line 8 (line 308): correct H-5 to H-5’
  • Page 13, line 315: How do you explain doublet for H-2’ (fructose moiety) of compound 12? I would also expect coupling with CH2 of CH2OH group and not only to H-3’ (fructose moiety), thus giving doublet of doublet?
  • Page 15, Table 2, compound 12: correct H-2’’ to H-2’
  • Page 16, line 370: correct “to be bind” to “to bind”
  • Page 20, Figure 10: I would rather use “Predicted 2D binding interactions” since this is based solely on the in silico method and not crystallographic data.
  • Page 23, line 511-512: How can you tell that they showed significant inhibition in extract, since many other compounds are present in extracts and the activity is not solely based on these two compounds?

Author Response

We agree with the general thrust of the reviewer’ comments – that there were many opportunities to improve the originally submitted manuscript – and have spent considerable time rewriting our original draft to address their concerns. As demonstrated by the responses below, we have carefully considered the reviewer’ points and addressed them by extensively rewriting the manuscript to improve its flow, coherence, and scientific content. All revised parts were marked with a yellow background in the manuscript. We are confident that the revised and restructured manuscript does not suffer from the problems identified by the reviewer in the original draft, and hope you will find it worthy for publication.

Comments of Reviewer 2

Responses

Tanzina Sharmin Nipun and co-authors presented a study about alpha-glucosidase inhibitors from Psychotria malayana jack leaves identifying ten putative bioactive compounds in this plant for the first time. In my opinion, the study itself is interesting for scientific community and the readers of Pharmaceuticals and could be published in this journal. Below you can find some of my corrections/ comments / suggestion to improve this manuscript:

Page 6: Concerning the identification of bioactive compounds: If you compare your chromatogram in Figure 4 to chromatogram of the pure compounds 1-9 (for example purchased standards) using the same methods, this would definitely confirm the presence of tested compounds in malayana leaves extracts if retention times match.

Please explain more in details how did you annotate the bioactive compounds – based on what (retention times using the same method as in National Institute of Standards database?)?

Amendment has been made as per suggestion in section 1: line 95-99; section 2.2.1: line 188-195, and section 4.6: line 782-788. The fragment m/z spectra of each putative compound are shown in Figure S3 – S10.

Page 13, 2.3.2. Identification of Putative Bioactive and Other Metabolites, line 8 (line 308): correct H-5 to H-5’

Amendment has been made as per suggestion (see section 2.3.2: line 338).

Page 13, line 315: How do you explain doublet for H-2’ (fructose moiety) of compound 12? I would also expect coupling with CH2 of CH2OH group and not only to H-3’ (fructose moiety), thus giving doublet of doublet?

We agree on the comment of the reviewer that H-2′ should not be doublet. After rechecking, we found that the chemical shift of H-2′ should be around 3.9ppm, which is in the crowded/overlapped area, and the signal cannot be seen clearly. Thus, we decide to remove the H-2′ assignment throughout the article (see section 2.3.2: line 344-345).

Page 15, Table 2, compound 12: correct H-2’’ to H-2’

Since the result related to H-2′ has been removed, then the amendment cannot be followed accordingly.

Page 16, line 370: correct “to be bind” to “to bind”

Amendment has been made as per suggestion (see section 2.4: line 392).

Page 20, Figure 10: I would rather use “Predicted 2D binding interactions” since this is based solely on the in silico method and not crystallographic data.

Amendment has been made as per suggestion (see Figure 10).

Page 23, line 511-512: How can you tell that they showed significant inhibition in extract, since many other compounds are present in extracts and the activity is not solely based on these two compounds?

We agree on the reviewer comment. Therefore, amendment has been made accordingly (see section 3.1: line 531-532).

Reviewer 3 Report

I found the manuscript “GC-MS- and NMR-Based Metabolomics and Molecular Docking Reveals the Potential Alpha-Glucosidase Inhibitors from Psychotria malayana Jack Leaves” interesting. Some observations are listed below:

A detailed methodology for Multivariate Data Analysis could be useful at the methods section or as a supplementary material to have a clearer idea on the results interpretation.

Page 6, line 177: “Various groups of bioactive compounds” refers to bioactive compounds against AG or bioactive compounds against different targets? Be careful by using the term bioactive metabolites, it can be interpreted as bioactive against AG. Also, a bioactive extract does not imply that all detected metabolites are responsible of the activity.

The Table 8 is very extensive and repetitive as compared to Fig 10, perhaps Table 8 can be included as a supplementary table. Also, the interactions in Table 8 and Fig 10 for the crystallographic ligand ADG and Quercetin can be included.

Line 502: The AGI activity of this compound is for the first time reported in this study. This sentence refers to compound 3, ¿its AGI activity is really reported? Or is only a component of the methanolic extract, but not necessarily have a contribution to the observed effect.

lines 838 to 841:  The conclusion “The detected metabolites, namely 1,3,5-benzenetriol, palmitic acid, cholesta-7,9(11)-diene-3-ol, 1-monopalmitin, β-tocopherol, α-tocopherol, 24-epicampesterol, stigmast-5-ene, 4-hydroxyphenylpyruvic acid, and glutamine, exhibited strong AGI activity” is not supported. Perhaps is better “The methanolic extract which contains the detected metabolites, namely 1,3,5-benzenetriol, palmitic acid, cholesta-7,9(11)-diene-3-ol, 1-monopalmitin, β-tocopherol, α-tocopherol, 24-epicampesterol, stigmast-5-ene, 4-hydroxyphenylpyruvic acid, and glutamine, exhibited strong AGI activity.

Author Response

We agree with the general thrust of the reviewer’ comments – that there were many opportunities to improve the originally submitted manuscript – and have spent considerable time rewriting our original draft to address their concerns. As demonstrated by the responses below, we have carefully considered the reviewer’ points and addressed them by extensively rewriting the manuscript to improve its flow, coherence, and scientific content. All revised parts were marked with a yellow background in the manuscript. We are confident that the revised and restructured manuscript does not suffer from the problems identified by the reviewer in the original draft, and hope you will find it worthy for publication.

Comments of Reviewer 3

Responses

I found the manuscript “GC-MS- and NMR-Based Metabolomics and Molecular Docking Reveals the Potential Alpha-Glucosidase Inhibitors from Psychotria malayana Jack Leaves” interesting. Some observations are listed below:

A detailed methodology for Multivariate Data Analysis could be useful at the methods section or as a supplementary material to have a clearer idea on the results interpretation.

Amendment has been made as per suggestion.

Detailed methodology of multivariate data analysis has been described in section 4.6: line 782-788, and section 4.9: line 827-831.

Supplementary materials have been provided as per suggestion (see Figure S1, S2, and S11).

Page 6, line 177: “Various groups of bioactive compounds” refers to bioactive compounds against AG or bioactive compounds against different targets?

Be careful by using the term bioactive metabolites, it can be interpreted as bioactive against AG. Also, a bioactive extract does not imply that all detected metabolites are responsible of the activity.

Amendment has been made accordingly. The term “bioactive compounds” is changed to “compounds possessing AGI activity” (see section 2.2.2:  line 203).

The Table 8 is very extensive and repetitive as compared to Fig 10, perhaps Table 8 can be included as a supplementary table.

Amendment has been made as per suggestion (Table S1)

Also, the interactions in Table 8 and Fig 10 for the crystallographic ligand ADG and Quercetin can be included.

Amendment has been made per suggestion (Table S1 and Figure 10)

Line 502: The AGI activity of this compound is for the first time reported in this study. This sentence refers to compound 3, its AGI activity is really reported? Or is only a component of the methanolic extract, but not necessarily have a contribution to the observed effect.

Our claim that compound 3 contributes to the AGI activity based on the results from the loading scatter plot (Figure 2B). In this plot, it was found that this compound located nearby the AGI activity, indicating the positive correlation to the AGI activity (as mentioned in section 2.2.1: line 188-191). This finding was also supported by the result of the in-silico study.

lines 838 to 841:  The conclusion “The detected metabolites, namely 1,3,5-benzenetriol, palmitic acid, cholesta-7,9(11)-diene-3-ol, 1-monopalmitin, β-tocopherol, α-tocopherol, 24-epicampesterol, stigmast-5-ene, 4-hydroxyphenylpyruvic acid, and glutamine, exhibited strong AGI activity” is not supported. Perhaps is better “The methanolic extract which contains the detected metabolites, namely 1,3,5-benzenetriol, palmitic acid, cholesta-7,9(11)-diene-3-ol, 1-monopalmitin, β-tocopherol, α-tocopherol, 24-epicampesterol, stigmast-5-ene, 4-hydroxyphenylpyruvic acid, and glutamine, exhibited strong AGI activity.

Amendment has been made as per suggestion (see section 5: line 882-885).